# MOOSE-Star: Unlocking Tractable Training for Scientific Discovery by Breaking the Complexity Barrier

**Zonglin Yang** [1]   **Lidong Bing** [1]

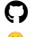   https://github.com/ZonglinY/MOOSE-Star
🤗   https://huggingface.co/collections/ZonglinY/MOOSE-Star

## Abstract

While large language models (LLMs) show promise in scientific discovery, existing research focuses on inference or feedback-driven training, leaving the direct modeling of the generative reasoning process, $P(\text{hypothesis}|\text{background})$ ($P(h|b)$), unexplored. We demonstrate that directly training $P(h|b)$ is mathematically intractable due to the combinatorial complexity ($O(N^k)$) inherent in retrieving and composing inspirations from a vast knowledge base. To break this barrier, we introduce MOOSE-Star, a unified framework that enables tractable and scalable training of $P(h|b)$, while supporting more scalable inference. In the best case, MOOSE-Star reduces complexity from exponential to logarithmic ($O(\log N)$) by (1) training on decomposed subtasks derived from the probabilistic equation of discovery, (2) employing motivation-guided hierarchical search to enable logarithmic retrieval and prune irrelevant subspaces, and (3) utilizing bounded composition for robustness against retrieval noise. To facilitate this, we release TOMATO-Star, a dataset of 108,717 decomposed papers (38,400 GPU hours) for training. Empirically, MOOSE-Star scales continuously with training data and inference budget, whereas direct brute-force sampling hits a "complexity wall."

## 1. Introduction

Recently, LLMs for scientific discovery have attracted increasing research attention. However, most research efforts

[1]MiroMind AI. Correspondence to: Zonglin Yang <zonglin.yang@miromind.ai>, Lidong Bing <lidong.bing@miromind.ai>.

*Proceedings of the 43rd International Conference on Machine Learning*, Seoul, South Korea. PMLR 306, 2026. Copyright 2026 by the author(s).

to date focus on developing inference methodologies to leverage LLMs for discovery, with only a few works investigating how to *train* an LLM specifically for this task (Luo et al., 2025). Critically, nearly all of these existing training approaches rely on external feedback to refine generated hypotheses, rather than explicitly modeling the core conditional probability $P(\text{hypothesis}|\text{background})$ (denoted as $P(h|b)$), where the background $b$ comprises the research question and the survey of prior literature. In essence, these methods learn how to *update* a hypothesis given feedback, but overlook the fundamental reasoning process required to *generate* a high-quality hypothesis directly from the research background. For example, Weng et al. (2024); Li et al. (2024) leverage open-access reviews to train critique models; Behzadifar et al. (2025) utilize alignment with observed data as reward; and Goel et al. (2025) derive rewards from goal-specific rubric-based LLM feedback.

Consequently, how to train $P(h|b)$ remains unknown. In this paper, we provide a theoretical analysis demonstrating why directly training $P(h|b)$ is intractable due to its inherent combinatorial complexity. Furthermore, we propose a recipe to overcome this barrier, enabling the tractable and, crucially, *scalable* training of $P(h|b)$.

To understand this intractability, we build upon the formalization established in prior studies (Yang et al., 2025b; Liu et al., 2025b), viewing the generation of a hypothesis $h$ as a composition of the research background $b$ and a sequence of latent *inspirations* retrieved from the global knowledge base. For instance, the discovery of Backpropagation can be viewed as composing the background of "multilayer logistic regression" with the inspiration of the "chain rule" from calculus. This compositional view is supported by cognitive science studies (Koestler, 1964; Benedek et al., 2012) and recent inference-based discovery methods (Yang et al., 2024; Wang et al., 2024b; Garikaparthi et al., 2025; Kumar et al., 2025). However, directly training $P(h|b)$ implies implicitly learning to retrieve the correct sequence of $k$ inspirations from a literature space of size $N$ (e.g., $N \approx 10^7$, encoded within the LLM's pre-trained parameters). This results in a

combinatorial search space of $O(N^k)$, rendering end-to-end training mathematically intractable.

To break this complexity barrier, we introduce MOOSE-Star (MS), a framework that transforms the intractable objective into a solvable form through a series of theoretical decompositions. Specifically, we leverage the mathematical formulation proposed by Yang et al. (2025b), which decomposes the combinatorial complexity of $P(h|b)$ into a set of subtasks with linear complexity: retrieving inspirations (sequentially) and composing each inspiration into a hypothesis. Crucially, while Yang et al. (2025b) utilized this equation solely for inference, we operationalize it here for training. This decouples the exponential $O(N^k)$ dependency into $k$ tractable sequential steps, reducing the sample complexity to a linear $O(k \times N)$.

However, even with this decomposition, explicitly scanning the full literature ($N$) remains computationally prohibitive. To address this, we introduce three key innovations: (1) **Hierarchical Search**: We organize the global literature into a semantic search tree, replacing the expensive linear scan with top-down navigation. This reduces the retrieval complexity from linear $O(N)$ to logarithmic $O(\log N)$ in the best-case scenario. (2) **Bounded Composition**: To handle retrieval imperfections, we introduce a semantic tolerance radius. By training the composition module on noisy inspirations within a "bounded window" centered on the exact inspiration $i^*$, we ensure the model is robust to inexact retrievals. (3) **Motivation Planning**: We demonstrate that explicitly modeling a "Motivation" variable—appended to the research background—acts as a dynamic generative root for the search tree. By effectively pruning irrelevant branches during hierarchical search, this reduces the number of search steps required to locate the relevant semantic subspace ($N_m < N$).

To support this data-hungry training paradigm, we developed a comprehensive pipeline that processes scientific literature across domains such as biology, chemistry, medicine, medical imaging, psychology, and cognitive science. In total, we extracted research backgrounds, ground-truth hypotheses, and inspirations—each linked to a historical citation—from 108,717 papers, a massive computational effort consuming approximately 38,400 A800 GPU hours. We release this processed dataset as the TOMATO-Star dataset.

Finally, we demonstrate in § 7 that MOOSE-Star enables scalable training while achieving superior test-time scaling over brute-force sampling. The decomposed discovery subtasks scale with increasing training data; at inference time, while brute-force sampling hits a complexity wall on hypotheses requiring multiple inspirations, our inspiration-based hierarchical search exhibits continuous improvement with increased budget, effectively turning an intractable discovery problem into a manageable search process.

Our contributions can be summarized as follows:

- We establish the first theoretical analysis on why training $P(h|b)$ is intractable (combinatorial complexity).

- We provide the first training recipe on how to enable the tractable and scalable training of $P(h|b)$.

- We provide an inference recipe that supports more scalable test-time inference for discovery.

- We release a dataset of 108,717 processed papers (cost 38,400 GPU hours), along with the full training & inference codebase and trained models.

## 2. Related Work

While a significant body of research focuses on novel inference algorithms for LLM-based discovery (Romera-Paredes et al., 2024; Novikov et al., 2025; Shojaee et al., 2025; Lu et al., 2024), few studies investigate methods for *training* LLMs for discovery. Most existing training efforts focus on constructing external feedback mechanisms for generated hypotheses rather than modeling the core conditional probability $P(h|b)$ itself. For instance, Weng et al. (2024); Li et al. (2024) leverage peer review data to train critique models, which serve as reward signals for the discovery agent. Similarly, in the domain of equation discovery, Behzadifar et al. (2025) derive rewards based on the goodness of fit between the proposed hypothesis (equation) and observed data. Goel et al. (2025) extract evaluation criteria from scientific literature to construct rubric-based LLM self-evaluators, while Pu et al. (2025) train models to retrieve ancillary high-level guidance from a constrained set to assist discovery. O'Neill et al. (2025) construct paper-derived supervision for training hypothesis generators; however, obtaining valid reasoning traces toward $h$ from $b$ alone remains difficult at scale (as discussed in § 7.1), and simulated reasoning traces generated directly from $b$ and the ground-truth $h$ are often difficult to make effective for training (Wang et al., 2025).

## 3. Preliminary: The Decomposition Theory

Our framework builds upon the probabilistic decomposition theory introduced in MOOSE-Chem (Yang et al., 2025b). Fundamentally, this theory posits that a scientific hypothesis $h$ is not generated *ex nihilo* (out of nothing) but is synthesized by composing a research background $b$ with a sequence of $k$ latent *inspirations*, $\mathbf{i} = (i_1, \dots, i_k)$, retrieved from a global knowledge base $\mathcal{I}$ ($N = |\mathcal{I}|$). Formally, the generation of a hypothesis can be modeled as:

$$h = f(b, i_1, \dots, i_k) \tag{1}$$

This compositional view is well-supported by cognitive science literature (Koestler, 1964) and has proven empirically

effective across scientific disciplines (Liu et al., 2025b). For instance, the discovery of *Backpropagation* can be viewed as composing the background of "multilayer logistic regression" with the inspiration of the "chain rule" from calculus; similarly, a new synthesis method in chemistry might be derived by applying a specific "catalyst" (inspiration) to a known reaction mechanism (background).

Based on this formulation, we can approximate the intractable marginal likelihood $P(h \mid b)$ by decomposing the generation process into $k$ sequential steps. Assuming a Markov property where the intermediate hypothesis $h_{j-1}$ sufficiently encapsulates the history of prior inspirations, the probability can be factorized using the chain rule as:

$$P(h \mid b) \approx \prod_{j=1}^{k} \underbrace{P(i_j \mid b, h_{j-1}, \mathcal{I})}_{\text{Inspiration Retrieval}} \cdot \underbrace{P(h_j \mid b, h_{j-1}, i_j)}_{\text{Hypothesis Composition}}$$

(2)

where $h_j$ denotes the intermediate hypothesis state after the $j$-th step (with $h_0 = b$ and $h_k = h$). The first term represents the *retrieval* of the next relevant inspiration, and the second term represents the *composition* of the hypothesis using the retrieved inspiration.

## 4. Dataset Construction

To enable the training of our framework, we construct TOMATO-Star, a large-scale dataset of scientific papers processed into structured components. The construction pipeline consists of four stages: collection, decomposition, structured representation, and quality assurance.

**Data Collection and Split.** We collected 108,717 open-access papers from the NCBI database (Schoch et al., 2020), spanning biology, chemistry, medicine, medical imaging, psychology, and cognitive science. The corpus covers the period from January 2020 to October 2025. To rigorously evaluate generalization, we enforce a strict temporal split: papers published from January 2020 to September 2025 constitute the training set, while papers from October 2025 serve as the held-out test set. We verify that the splits do not overlap. October 2025 is sufficiently after the knowledge cutoff of the R1-Distilled-Qwen models used in our experiments to substantially reduce contamination risk.

**Preprocessing and Decomposition.** Raw PDF documents are first converted to Markdown format using MinerU (Wang et al., 2024a). We then employ a pipeline of locally deployed reasoning models, specifically DeepSeek-R1 and R1-distilled-Qwen-32b (Guo et al., 2025), to decompose each paper into a tuple of $(b, h, \mathbf{i})$: *Research Background*, *Hypothesis*, and *Inspirations*. This workflow adapts the design of ResearchBench (Liu et al., 2025b) but introduces stricter quality controls and a novel hypothesis representation scheme.

**Structured Representation.** The decomposed components are defined as follows:

- **Research Background** ($b$)**:** Comprises the specific research question and a background survey summarizing prior methods addressing that question.

- **Inspirations (i):** Ground-truth inspirations are identified directly from the source paper's citations. We augment these citations by retrieving their full titles and abstracts via Semantic Scholar (Kinney et al., 2023).

- **Hypothesis** ($h$) **as Incremental Deltas:** Unlike standard summaries, we structure the hypothesis as a sequence of "Delta Hypotheses" ($\Delta h$), enforcing a one-to-one mapping where each inspiration $i_j$ results in exactly one $\Delta h_j$. The final hypothesis is the concatenation of these deltas. Crucially, each $\Delta h$ is structured into three levels: (1) *Motivation* (why this direction was chosen), (2) *Mechanism* (why it can work), and (3) *Methodology* (how it is implemented).

**Quality Assurance.** To ensure data integrity, every processed sample must pass four automated quality checks prior to inclusion: (1) **Information Necessity:** Each proposed $i$ must provide essential, complementary knowledge required to derive $h$ from $b$. (2) **Information Sufficiency:** The integration of the background and inspirations must logically entail the hypothesis ($b + \mathbf{i} \approx h$). (3) **Information Disjointness:** The background $b$ must remain strictly independent, leaking no information present in either the inspirations or the hypothesis. (4) **Non-Redundancy:** The extracted inspirations within $\mathbf{i}$ must be mutually distinct and non-repetitive.

## 5. Methodology

In this section, we first analyze *why* directly training $P(h|b)$ is computationally intractable. We then derive the MOOSE-Star framework, showing how each component progressively transforms the intractable objective into a solvable problem with manageable complexity.

### 5.1. Analysis: The Intractability Barrier ($O(N^k)$)

Consider the standard baseline approach: training a model to directly maximize the marginal likelihood $P(h \mid b)$ end-to-end. As established in Eq. 1, generating a valid hypothesis inherently requires identifying and synthesizing a sequence of $k$ inspirations $\{i_1, \ldots, i_k\}$. Consequently, direct modeling $P(h \mid b)$ implies an implicit search over the Cartesian product of the knowledge base, $\mathcal{I}^k$. Even disregarding the computational cost of composition, the search space complexity alone scales exponentially:

$$\mathcal{C}_{\text{E2E}} > O(N^k) \quad (\text{e.g., for } N = 10^7, k = 3 :\approx 10^{21}) \quad (3)$$

Here, $N = |\mathcal{I}| \approx 10^7$ represents the scale of global scientific literature, where each paper constitutes an independent unit of knowledge. Crucially, this complexity barrier persists even in the absence of an external retriever. When relying solely on internal weights, the LLM is forced to implicitly retrieve from the knowledge encoded during pre-training. Given this intractable $O(N^k)$ search space, standard end-to-end training becomes mathematically ill-posed, resulting in severe convergence difficulties.

## 5.2. Method I: Decomposed Sequential Training ($O(N)$)

To dismantle the exponential barrier of $O(N^k)$, our first methodology shifts from directly training the monolithic $P(h \mid b)$ to translating the theoretical decomposition in Eq. 2 into a concrete training objective. We train the model on two equivalent sequential subtasks: *Inspiration Retrieval* and *Hypothesis Composition*.

For each step $j$, the complexity breakdown is as follows:

- **Inspiration Retrieval (IR):** The task is to identify the correct ground-truth inspiration from the full global knowledge base. This requires scanning the search space of size $N$, resulting in a complexity of $O(N)$.

- **Hypothesis Composition (HC):** The task is to generate $h_j$ given the retrieved inspiration $i_j$. Conditioned on the provision of the exact ground-truth inspiration, the generation in the ideal case requires only a single inference pass. Thus, relative to the search space size $N$, this step has a constant complexity of $O(1)$.

Since this process is repeated sequentially for $k$ steps, this methodology fundamentally transforms the search space topology from a Cartesian product ($N^k$) to a linear sum ($k \times N$). Consequently, the complexity is:

$$\mathcal{C}_\text{I} \approx k \times \left[ \underbrace{O(N)}_{\text{Inspiration Retrieval}} + \underbrace{O(1)}_{\text{Hypothesis Composition}} \right] \quad (4)$$

We formulate both IR and HC as generative reasoning tasks, requiring the model to produce a Chain-of-Thought (CoT) (Wei et al., 2022) rationale before outputting the final answer. To optimize this reasoning capability, we adopt teacher-based Rejection Sampling Fine-Tuning (RFT) (Touvron et al., 2023; Zelikman et al., 2022).

For the IR task, the model is trained to identify the target inspiration via a generative selection process. The input consists of the query context and a pool of 15 papers (1 positive and 14 negatives), each represented by its title and abstract. The negative set comprises *easy negatives* (randomly sampled papers published no later than the source paper) and *hard negatives* drawn from three categories: papers with key-word overlap with the source background, papers with keyword overlap with the ground-truth inspiration, and papers semantically similar to the ground-truth inspiration. We acknowledge that these negatives might occasionally indicate valid alternative inspirations. However, the ground-truth citation represents the author's explicit acknowledgment of causal influence, providing the most reliable supervision for prioritizing optimal inspirations over merely related ones. Furthermore, this single-selection objective minimizes the penalty for potentially valid unlabelled candidates compared to independently labeling each candidate.

For the HC task, the objective is to generate the incremental hypothesis update, $\Delta h_j$. The model is conditioned on the research background $b$, the retrieved ground-truth inspiration $i_j$, and $h_{j-1}$, represented by the concatenation of $\{\Delta h_1, \dots, \Delta h_{j-1}\}$ (if $j \geq 2$). To enable effective rejection sampling, we developed a rubric-based evaluator to compare the quality of sampled hypotheses against the ground-truth $\Delta h_j^*$, ensuring that only high-quality samples are retained for training. The evaluation rubric is provided in § H.

## 5.3. Method II: Bounded Composition

While the linear decomposition $O(kN)$ breaks the exponential barrier, linearly scanning $N \approx 10^7$ papers for every retrieval step remains computationally prohibitive for test-time inference. To address this, we must further optimize the linear term $O(N)$.

We achieve this by relaxing the retrieval objective. In § 5.2, the formulation assumes a strict "Exact Match" constraint: the IR module must identify the unique ground-truth inspiration $i^*$ (1 out of $N$), and the HC module relies solely on this exact input. We propose **Bounded Composition**, which introduces a **Semantic Tolerance Space** of size $M$. Instead of requiring $i^*$, we posit that the HC module can be trained to function robustly within a semantic neighborhood $\mathcal{I}_{i^*}$ centered on $i^*$ containing $M$ candidates (where $i^* \in \mathcal{I}_{i^*} \subset \mathcal{I}$). If the retriever provides any inspiration $i \in \mathcal{I}_{i^*}$, the HC module is expected to reason through this proxy $i$ to recover the underlying $i^*$ and correctly compose the hypothesis. This relaxation alters the complexity landscape:

- **IR Relaxation ($O(N) \rightarrow O(N/M)$):** The retrieval target expands from a single point to a subspace of size $M$. In a randomized search or sampling process, the expected number of steps to hit a target of size $M$ in a space of $N$ reduces linearly to $N/M$.

- **HC Overhead ($O(1) \rightarrow O(M)$):** The composition cost increases. The HC model must now perform additional reasoning to resolve the ambiguity within the tolerance space. We represent this abstractly as an internal complexity of $O(M)$, reflecting the cost of reasoning within the local space of $M$ possibilities.

Consequently, the overall complexity updates to:

$$\mathcal{C}_{\{I,II\}} \approx k \times \left[ \underbrace{O\left(\frac{N}{M}\right)}_{\text{Relaxed Retrieval}} + \underbrace{O(M)}_{\text{Bounded Composition}} \right] \quad (5)$$

This trade-off is strategic: while HC complexity increases linearly with $M$, the dominant IR complexity decreases by a factor of $M$. Given that $N \gg M$, shifting the computational burden from the global search space to local reasoning is a significant net reduction in total complexity.

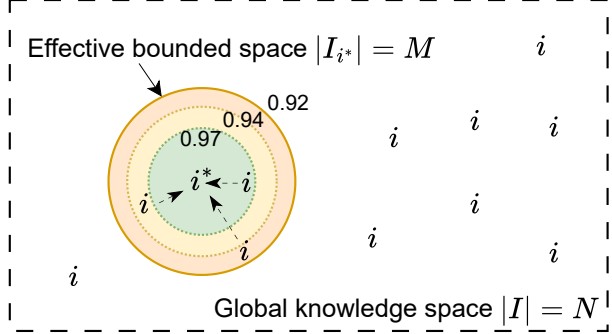

*Figure 1.* Bounded Composition. The concentric circles around $i^*$ represent similarity thresholds that define the bounded space $M$.

To operationalize Bounded Composition, we first curate a pool of semantic proxies. For each ground-truth inspiration $i^*$ in the dataset, we query Semantic Scholar to retrieve up to 50 semantically similar papers. We embed these candidates using SPECTER2 (Singh et al., 2022) and compute their cosine similarity to $i^*$. Based on these scores, we stratify valid candidates into three difficulty tiers: *Easy* ($[0.94, 0.97)$), *Medium* ($[0.92, 0.94)$), and *Hard* ($[0.90, 0.92)$). From each available tier, we select the single candidate with the highest similarity score to serve as the tier representative. We then employ the teacher model to generate hypotheses conditioned on these proxy inspirations rather than the exact ground truth. The generated outputs are rigorously evaluated against the quality control rubric described in § H. Crucially, for instances where proxies from multiple tiers yield passing hypotheses, we prioritize the sample from the most difficult tier (i.e., lowest similarity) for the final training set. This strategy explicitly maximizes the learned semantic tolerance space, forcing the model to reason robustly even with less precise retrieval contexts.

### 5.4. Method III: Hierarchical Search ($O(\log N)$)

Despite the reduction to $O(N/M)$, a linear scan over the global knowledge base $\mathcal{I}$ remains inefficient. To enable logarithmic-time search (at least in the best-case scenario), we replace the flat scan with a **Hierarchical Best-First Search**. This design is inspired by MOOSE-Chem2 (Yang et al., 2025a), which formally introduces the fine-grained hypothesis discovery task, formulates it as an optimization problem, and demonstrates that this optimization process can be made more tractable by organizing it hierarchically over multiple abstraction levels. While MOOSE-Chem2 develops this principle in the hypothesis space, here we adapt it to inspiration retrieval by constructing the hierarchy directly over the inspiration space $\mathcal{I}$.

**Tree Construction (Offline).** We construct the search tree via a bottom-up recursive clustering approach. First, all papers in $\mathcal{I}$ are embedded using SPECTER2. We then perform hierarchical K-means clustering (McQueen, 1967) recursively from the leaves up to the root. In each iteration, we cluster the current level's embeddings (initially the papers, subsequently the cluster centroids), where the resulting centroids serve as the nodes for the next higher level of the hierarchy. This process repeats until a single root centroid remains. Crucially, to align with our IR model's capacity, we enforce a maximum branching factor of $c = 15$. We employ a post-clustering **balancing operation**: if a cluster exceeds size $c$, its furthest member (least similar to the centroid) is reassigned to its nearest centroid in the same hierarchy that has available capacity (size $< c$). This ensures the tree is balanced with a consistent branching factor.

**Best-First Search (Online).** During inference, we traverse the tree using a Best-First Search strategy. We maintain a *priority queue* of candidate nodes, ranked by a path confidence score. In each iteration, we pop the highest-scoring node for expansion. The IR model evaluates the sub-branches of this node, generating a probability distribution over its children (derived from the output token log-probabilities). Crucially, these local probabilities are aggregated with the parent node's stored cumulative probability to compute updated scores for each child, which are then pushed back into the priority queue. To ensure fair comparison between nodes at different depths—avoiding the bias against deeper paths inherent in raw cumulative products—we define the ranking score as the **geometric mean** of the probabilities along the path from the root:

$$\text{Score}(path_j) = \sqrt[j+1]{\prod_{t=0}^{j} p_t} \quad (6)$$

where $p_t$ is the probability assigned by the IR model at depth $t$. By dynamically updating the search frontier based on this length-normalized metric, the algorithm effectively prioritizes the most promising semantic neighborhoods without traversing the entire database.

In the best-case scenario (where the IR model makes ideal routing decisions), the search complexity corresponds to the depth of the tree, which is logarithmic with respect to the

search space. This yields more tractable complexity:

$$\mathcal{C}_{\{I,II,III\}} \approx k \times \Bigg[ \underbrace{O\left(\log \frac{N}{M}\right)}_{\text{Hierarchical Search}} + \underbrace{O(M)}_{\text{Bounded Composition}} \Bigg] \quad (7)$$

### 5.5. Method IV: Motivation Planning

While hierarchical search optimizes the traversal of the knowledge base, it is essentially an unguided navigation: the search process often lacks explicit direction, essentially "stumbling" into candidates without prior intent. To address this, we introduce a **Motivation** variable ($m$).

**Semantic Guidance.** Functionally, $m$ serves as a **Directional Guide** generated prior to retrieval. By appending $m$ to the query context $b$, we explicitly condition the hierarchical search on specific intended properties or requirements. This *biases* the search focus towards relevant semantic manifolds, filtering out branches inconsistent with the high-level intent. Consequently, the effective search space is reduced from the global knowledge base $N$ to a motivation-aligned subspace $N_m$ (where $N_m < N$).

**O(1) Generation.** Crucially, the motivation $m$ is derived solely from the research background $b$ via internal abstraction (e.g., a single-step translation of background information into high-level requirements). We intentionally design this planning step to be lightweight—approximately $O(1)$ (a single inference)—to ensure it neither becomes a computational bottleneck nor generates overly specific directives that could misguide the subsequent computationally expensive IR and HC resources. Assuming the generated motivation is sound (a reasonable assumption for this lightweight translation step), the overall complexity is further reduced to:

$$\mathcal{C}_{\text{MS}} \approx k \times \Bigg[ \underbrace{O(1)}_{\text{Motivation}} + \underbrace{O\left(\log \frac{N_m}{M}\right)}_{\text{Focused Retrieval}} + \underbrace{O(M)}_{\text{Bounded Composition}} \Bigg] \quad (8)$$

**Hierarchical MDP Formulation.** This addition formally extends the decomposition in Eq. 2 into a tri-stage Hierarchical Markov Decision Process (HMDP). At each constructive iteration $j$, the model first formulates a high-level intent $m_j$ before executing retrieval and composition:

$$P(h \mid b) \approx \prod_{j=1}^{k} \Bigg[ \underbrace{P(m_j \mid b, h_{j-1})}_{\text{Motivation Planning}} \cdot \underbrace{P(i_j \mid b, h_{j-1}, m_j, \mathcal{I})}_{\text{Inspiration Retrieval}} \cdot \underbrace{P(h_j \mid b, h_{j-1}, m_j, i_j)}_{\text{Hypothesis Composition}} \Bigg] \quad (9)$$

The formal derivation of Eq. 9 is provided in § B. Compared to the base formulation, the additional term $P(m_j \mid b, h_{j-1})$ explicitly governs the motivation planning step, ensuring that the subsequent retrieval ($i_j$) and composition ($h_j$) remain aligned with a coherent research trajectory.

## 6. Experiment

In this section, we empirically evaluate the four methodologies proposed in § 5. All experiments utilize the TOMATO-Star test set, comprising papers published in October 2025, to ensure a rigorous evaluation free from contamination.

### 6.1. Method I: Decomposed Sequential Training

For both subtasks, we use R1-DISTILLED-QWEN-32B as the teacher to generate training samples, which are filtered to fine-tune the student model R1-DISTILLED-QWEN-7B.

*Table 1.* Inspiration retrieval results.

| Model | Accuracy |
|---|---|
| *Frontier Models* | |
| GPT-4O-MINI | 41.99% |
| CLAUDE-SONNET-4.6 | 45.02% |
| DEEPSEEK-R1 | 45.11% |
| GPT-4O | 48.37% |
| GEMINI-3-FLASH | 51.44% |
| GPT-5.4 | 51.50% |
| GEMINI-3-PRO | **54.89%** |
| *Baselines* | |
| Random Selection | 6.70% |
| R1-DISTILLED-QWEN-7B | 28.42% |
| MS-IR-7B | **54.37%** |
| MS-7B | 54.34% |

*Table 2.* Hypothesis composition performance given the ground-truth inspiration $i^*$, judged by GPT-4O under the *M3 rubric*. "Total" aggregates the scores for Motivation (Mot), Mechanism (Mec), and Methodology (Met). "w/ $n\times$ bounded" indicates the model was trained with $n$ additional copies of bounded composition data.

| Model | Total | Mot | Mec | Met | Length |
|---|---|---|---|---|---|
| *Frontier Models* | | | | | |
| GPT-4O-MINI | 5.18 | 2.26 | 1.64 | 1.28 | 175.29 |
| GPT-4O | 5.99 | 2.43 | 1.95 | 1.60 | 264.93 |
| DEEPSEEK-R1 | 6.42 | 2.54 | 2.10 | 1.77 | 264.46 |
| GEMINI-3-FLASH | 6.42 | 2.48 | 2.16 | 1.79 | 351.79 |
| GEMINI-3-PRO | 6.70 | 2.59 | 2.26 | 1.85 | 300.86 |
| CLAUDE-SONNET-4.6 | 7.28 | 2.78 | 2.41 | 2.08 | 510.26 |
| GPT-5.4 | **7.82** | **2.99** | **2.61** | **2.22** | 504.60 |
| *Baselines* | | | | | |
| R1-DISTILLED-QWEN-7B | 4.05 | 1.96 | 1.30 | 0.80 | 231.02 |
| MS-HC-7B | 4.68 | 2.13 | 1.46 | 1.09 | 204.12 |
| w/ $1\times$ bounded | **4.74** | **2.16** | **1.48** | **1.10** | 203.84 |
| w/ $2\times$ bounded | 4.73 | 2.15 | **1.48** | 1.09 | 205.17 |
| MS-7B | **5.02** | **2.22** | **1.59** | **1.20** | 208.98 |

*Table 3.* Hypothesis composition performance under inspiration noise, judged by GPT-4o under the *M3 rubric*. The tiers **Easy** (E), **Medium** (M), and **Hard** (H) correspond to the semantic similarity of the retrieved proxy inspiration to the ground truth. "w/ $n\times$ bounded" indicates the model was trained with $n$ additional copies of bounded composition data.

| Model | Easy (E) | | | | Medium (M) | | | | Hard (H) | | | |
|---|---|---|---|---|---|---|---|---|---|---|---|---|
| | Total | Mot | Mec | Met | Total | Mot | Mec | Met | Total | Mot | Mec | Met |
| R1-DISTILLED-QWEN-7B | 2.72 | 1.50 | 0.70 | 0.51 | 2.27 | 1.35 | 0.55 | 0.38 | 2.00 | 1.24 | 0.46 | 0.31 |
| MS-HC-7B | 3.10 | 1.63 | 0.81 | 0.66 | 2.59 | 1.46 | 0.61 | 0.53 | 2.39 | 1.38 | 0.54 | 0.48 |
| w/ $1\times$ bounded | 3.09 | 1.61 | 0.81 | 0.67 | 2.72 | 1.48 | **0.66** | 0.57 | 2.42 | 1.39 | 0.55 | 0.47 |
| w/ $2\times$ bounded | **3.18** | **1.67** | **0.83** | **0.68** | 2.74 | 1.51 | 0.66 | 0.57 | 2.56 | 1.45 | 0.58 | 0.53 |
| MS-7B | **3.37** | **1.72** | **0.90** | **0.75** | **2.86** | **1.55** | **0.69** | **0.62** | **2.78** | **1.52** | **0.66** | **0.60** |

**Inspiration Retrieval (IR).** We fine-tuned the student on 152k retrieval examples, resulting in MS-IR-7B (where MS stands for MOOSE-Star). As shown in Table 1, this specialized training yields a substantial performance gain, boosting accuracy from 28.42% to 54.37%. We verify in § C that this improvement reflects genuine retrieval ability rather than exploitation of superficial patterns.

**Hypothesis Composition (HC).** We fine-tuned the student on 97k samples generated via teacher-based rejection sampling, resulting in MS-HC-7B. Performance is evaluated using the *M3 rubric* (detailed in § H), an LLM-based rubric summing scores for *Motivation*, *Mechanism*, and *Methodology*. Each dimension ranges from [0, 4] (representing 0%–100% information recall), yielding a total score of 12. We use GPT-4o (OpenAI, 2024) as the judge model. Table 2 presents the results on ground-truth inputs. The fine-tuned MS-HC-7B significantly outperforms the baseline. Furthermore, results indicate that incorporating "bounded" training data (hypotheses generated from noisy inspirations) further enhances performance, suggesting that exposure to semantic variations during training improves reasoning robustness even when evaluating on perfect inspirations.

**Overall.** These results suggest an interesting asymmetry between the two subtasks. For inspiration retrieval, targeted post-training appears sufficient to elevate a 7B model to near-frontier performance. By contrast, although hypothesis composition improves substantially over the base model, it remains clearly below frontier LLMs. This pattern suggests that retrieval may rely more on activating and organizing latent scientific associations already present in the base model, whereas high-quality composition likely depends more strongly on the base model's underlying scientific reasoning and generation capacity.

Motivated by the complementary nature of the two tasks, we jointly train on both IR and HC data (w/ $1\times$ bounded), yielding a unified model MS-7B. As shown in Tables 1 and 2, MS-7B retains IR accuracy (54.34% vs. 54.37%) while improving HC performance (5.02 vs. 4.68), suggesting that multi-task training provides a beneficial inductive

bias for hypothesis composition without degrading retrieval.

## 6.2. Method II: Bounded Composition

Table 3 evaluates performance under noisy conditions, using proxy inspirations stratified by semantic similarity to $i^*$ (*Easy*, *Medium*, *Hard*). The results demonstrate that incorporating bounded training data significantly boosts M3 scores across all tiers compared to the base model. MS-7B further improves across all tiers, with the largest gain on *Hard* proxies (+0.22 total over w/ $2\times$ bounded), indicating that multi-task training strengthens robustness to retrieval noise. Similarly, GPT-4o is used as the judge model.

## 6.3. Method III: Hierarchical Search

*Table 4.* Efficiency and accuracy of hierarchical search variants compared to a tournament baseline. **IR inference calls**: Average number of IR model queries required to identify the ground-truth inspiration. **Proposed Rank**: The average position of the ground-truth inspiration in the retrieved list (lower is better).

| Search Method | IR inference calls | Proposed Rank |
|---|---|---|
| Tournament Search | 218.00 | 987.76 |
| Hierarchical Search | 67.78 | 813.40 |
| w/ motivation (simple) | 63.80 | 767.64 |
| w/ motivation (detailed) | **63.05** | **742.50** |

Using MS-IR-7B for routing, we evaluate search efficiency on a corpus of 3,035 ground-truth inspirations derived from the 1,658 test papers (October 2025). A hierarchical tree is constructed over this corpus as described in § 5.4. Table 4 compares our Hierarchical Best-First Search against a Tournament Search baseline. The baseline employs an exhaustive bottom-up strategy, processing all candidates through the cluster structure to identify winners, which incurs a high, fixed computational cost. We report two metrics: *IR Inference Calls* (the average number of IR queries required to successfully locate $i^*$) and *Proposed Rank* (the average position of $i^*$ in the retrieval order, where lower is better).

The results demonstrate that our hierarchical approach is significantly more efficient, reducing inference calls by ap-

proximately $3\times$ (67.78 vs. 218.00) while achieving a superior (lower) average rank. This confirms that the top-down, probability-guided search effectively prunes irrelevant branches without compromising retrieval accuracy.

### 6.4. Method IV: Motivation Planning

We analyze the impact of motivation quality on search efficiency using ground-truth-derived directives. We compare two variants: (1) **Simple**: A direct translation of requirements from $b$ with minimal reasoning and specification. (2) **Detailed**: Derived from the delta hypothesis, which we observe typically captures strategic intent without leaking mechanism or methodology details. As shown in Table 4, incorporating motivation consistently improves search efficiency. The detailed variant achieves the best performance, confirming that more detailed and precise planning can further reduce the computational effort required to locate $i^*$.

## 7. Scaling Analysis

### 7.1. Scaling Brute-force Sampling $P(h|b)$

*Table 5.* Per-attempt pass rates of brute-force end-to-end (BF) vs. decomposed subtasks (HC, IR), all generated by R1-DISTILLED-QWEN-32B. BF and HC are evaluated with the same model under the *M3 rubric* ($\geq 8/12$); IR is judged by retrieval correctness.

| $k$ | BF | HC | IR |
|---|---|---|---|
| 1 | 9.46% (1,674 / 17,692) | 18.88% (51,564 / (273,175 ×**1**)) | 27.84% (91,634 / (329,178 ×**1**)) |
| 2 | 1.17% (393 / 33,558) | 22.53% (208,723 / (463,230 ×**2**)) | 25.48% (295,455 / (579,769 ×**2**)) |
| 3 | 0.23% (20 / 8,683) | 25.75% (90,900 / (117,663 ×**3**)) | 24.22% (109,224 / (150,351 ×**3**)) |
| *Total* | *3.48%* (2,087 / 59,933) | *22.62%* (351,187 / 1,552,624) | *25.59%* (496,313 / 1,939,769) |

Direct distillation from the end-to-end distribution $P(h|b)$ via brute-force (BF) sampling is prohibitively inefficient due to the scarcity of valid reasoning traces. For a fair per-attempt comparison, all attempts in Table 5 are generated by R1-DISTILLED-QWEN-32B; BF and HC are scored by the same model under the M3 pass criterion (§ H), while IR is judged by retrieval correctness. A BF sample is counted as positive for RFT only if its single generated hypothesis satisfies the M3 pass criterion against all target deltas $\{\Delta h_1^*, \ldots, \Delta h_k^*\}$; an HC attempt is counted at the step level and judged only against its corresponding target $\Delta h_j^*$.

Table 5 shows that BF pass rates drop from $9.46\%$ at $k = 1$ to $1.17\%$ at $k = 2$ and $0.23\%$ at $k = 3$ — a $\sim41\times$ collapse, making brute-force RFT impractical under reasonable compute budgets. In contrast, decomposed subtasks remain approximately stable across $k$, with the mild HC drift analyzed in Appendix D. This empirically corroborates the complexity analysis in § 5: BF must jointly sample all $k$ innovations, so its success probability compounds with $k$, while decomposition reduces the problem to $k$ per-step decisions with bounded per-step success rates, corresponding to linear rather than exponential scaling with inspiration depth.

This gap reflects two structural advantages of the decomposition. **First**, HC targets a single increment $\Delta h_j$ rather than the joint distribution of all $k$ innovations. **Second**, conditioning on $i_j^*$ delegates search to IR and guides generation. Together, these factors break the training deadlock.

### 7.2. Scaling Training of MOOSE-Star

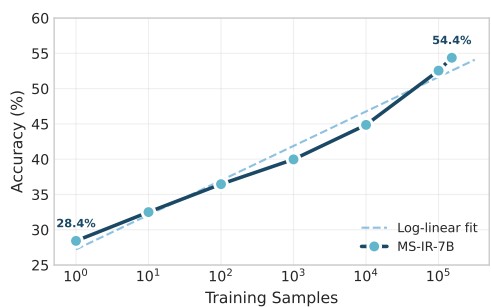

*(a)* Inspiration retrieval.

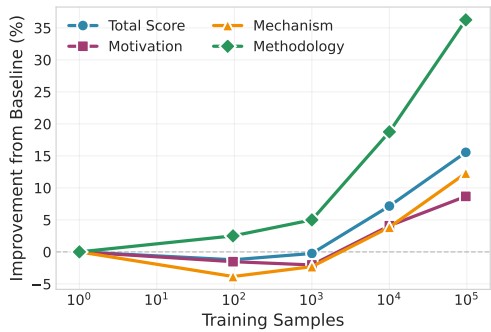

*(b)* Hypothesis composition.

*Figure 2.* Scaling laws of MOOSE-Star: (a) IR and (b) HC.

We analyze the scaling behavior of the dedicated IR and HC models (i.e., MS-IR-7B and MS-HC-7B, each trained on its respective subtask data) with training data ranging from $10^0$ to $10^5$ samples (Figure 2a and 2b). The IR model demonstrates a standard log-linear improvement in accuracy. In contrast, the HC model exhibits a distinct threshold behavior: significant log-linear gains emerge only after the dataset size exceeds $10^3$ samples, likely because HC is a generative task requiring higher data density than the classification-style IR task. Beyond confirming that the decomposition in Equation 2 makes scientific discovery *tractable*, these log-linear trends validate it as a *scalable* training paradigm.

**Generalization on OOD Retrieval.** The continuous scaling of the IR task is particularly noteworthy. Prior literature typically characterizes scientific inspiration retrieval as Out-of-Distribution (Yang et al., 2025b; Liu et al., 2025b), since true discovery requires associating concepts never linked in the prior knowledge base. Despite this novelty barrier, the consistent log-linear improvement suggests the model

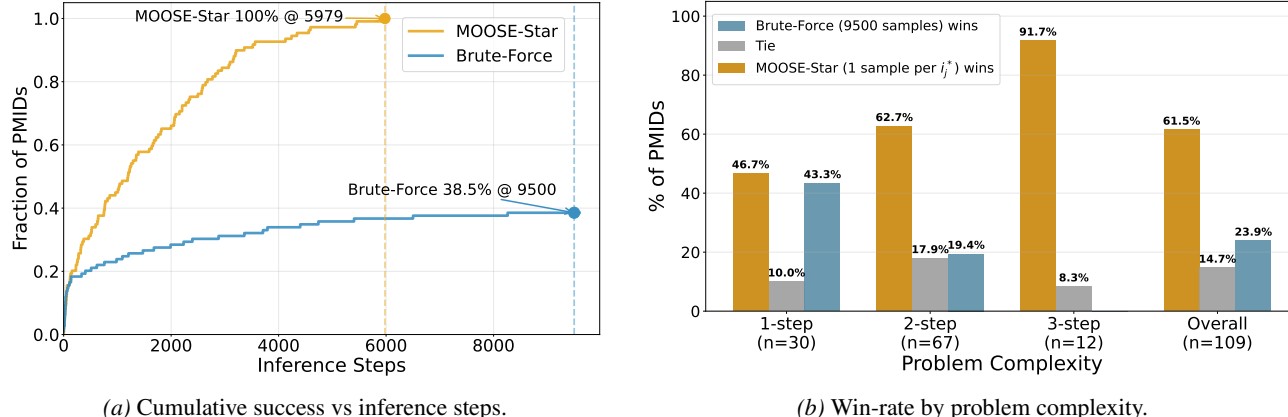

*(a)* Cumulative success vs inference steps.

*(b)* Win-rate by problem complexity.

*Figure 3.* Test-time scaling behavior of MOOSE-Star (MS-7B) versus brute-force sampling (R1-DISTILLED-QWEN-7B).

is not merely memorizing seen connections, but acquiring a generalizable "logic of discovery"—an abstract intuition for identifying meaningful scientific associations.

### 7.3. Test-time Scaling of MOOSE-Star

Figure 3a compares the test-time scaling of MOOSE-Star against a brute-force baseline on the first 109 test cases of TOMATO-Star, which together cover 200 sequential inspiration steps ($k \in [1, 3]$ per paper). At its core, this experiment asks: even with ∼9,500 brute-force samples, can it match what MOOSE-Star produces with only one HC composition per inspiration step (with $k \in [1, 3]$ steps per paper)?

**Cost (x-axis).** We measure computational cost as **cumulative inference calls**. For MOOSE-Star, this is the sum over all $k$ steps of (1) the IR calls to navigate the hierarchical search tree to identify $i_j^*$, and (2) HC calls equal to the **proposed rank** of $i_j^*$—since each retrieved inspiration triggers one HC composition. For brute force, the cost equals the number of end-to-end samples drawn from $P(h \mid b)$, each conditioned only on the research background.

**Success (y-axis).** We use MOOSE-Star as the comparison reference: conditioned on the ground-truth inspirations $i^*$, it serves as a high-quality proxy for $h^*$, and pairwise judgment is more robust than absolute scoring. To make the two methods directly comparable, we cast each side into a single hypothesis: brute force already produces one per sample, while we concatenate MOOSE-Star's $k$ step $\Delta h_j$ into a unified output. We then pair the MOOSE-Star hypothesis with each brute-force sample and ask Gemini-3-Flash (Google DeepMind, 2025) which one better captures the innovative essence of the ground-truth hypothesis $h^*$. Every pair is evaluated four times (two independent trials, each with original and swapped positions) to mitigate position bias and inference variance. Both curves share the cost axis. The brute-force curve at $x$ plots the fraction of papers where

at least one of the first $x$ samples wins or ties the pairwise comparison; the MOOSE-Star curve at $x$ plots the fraction whose pipeline has completed within $x$ calls.

**Results.** With a per-paper budget of ∼6,000 calls, MOOSE-Star completes all 109 papers, whereas brute force saturates at ∼38.5% after ∼9,500 samples; § E analyzes this saturation from a failure-probability perspective.

Figure 3b stratifies these outcomes by problem complexity $k$. On single-step tasks ($k = 1$), brute force achieves a non-trivial ∼53% win-or-tie rate, but this collapses with depth—to ∼37% at $k = 2$ and ∼8% at $k = 3$ (strict wins of 43%, 19%, and 0% respectively). Even nearly ten thousand samples cannot break the combinatorial complexity of recovering multiple novel concepts from $P(h \mid b)$ alone.

We provide an additional analysis on temporally held-out October 2025 papers in § F, including both quantitative joint retrieval–composition evaluation and a case study.

## 8. Conclusion

We address the intractability of training LLMs for scientific discovery by demonstrating that end-to-end modeling of $P(h|b)$ is ill-posed due to combinatorial complexity. To overcome this, we introduce MOOSE-Star, which operationalizes the probabilistic decomposition proposed in MOOSE-Chem to enable tractable and scalable training. Through Decomposed Sequential Training, Hierarchical Search, Bounded Composition, and Motivation Planning, MOOSE-Star reduces search complexity from exponential to logarithmic in the best case. Empirically, the decomposed subtasks exhibit log-linear performance scaling with increasing training data, and MOOSE-Star shows continuous test-time scaling where brute-force methods hit a "complexity wall" on hypotheses requiring multiple inspirations.

## Impact Statement

MOOSE-Star establishes a training-side foundation for LLM-based scientific discovery, complementing the predominantly inference-time methods of prior work. By making $P(h|b)$ tractably trainable, it opens scalable post-training as a research axis for discovery. The open release of TOMATO-Star (108,717 decomposed papers, approximately 38,400 GPU hours to construct) together with our trained models lowers the barrier for the broader community—including groups without comparable compute—to participate in this direction, much as shared open corpora have supported progress in adjacent areas of machine learning. Scalable training is one part of a broader research program toward discovery-capable LLMs; accessible interfaces that keep human scientists in the loop are an equally important complement (An & Yang, 2026).

Hypotheses generated by MOOSE-Star are proposals, not validated discoveries; in domains such as biology and medicine, acting on incorrect hypotheses can waste substantial experimental resources, and expert review and downstream validation remain indispensable. We therefore recommend that outputs of MOOSE-Star and similar systems be treated as candidate hypotheses for expert evaluation, and that downstream uses disclose the involvement of AI-generated reasoning.

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

## A. MOOSE-Star Concept Figure

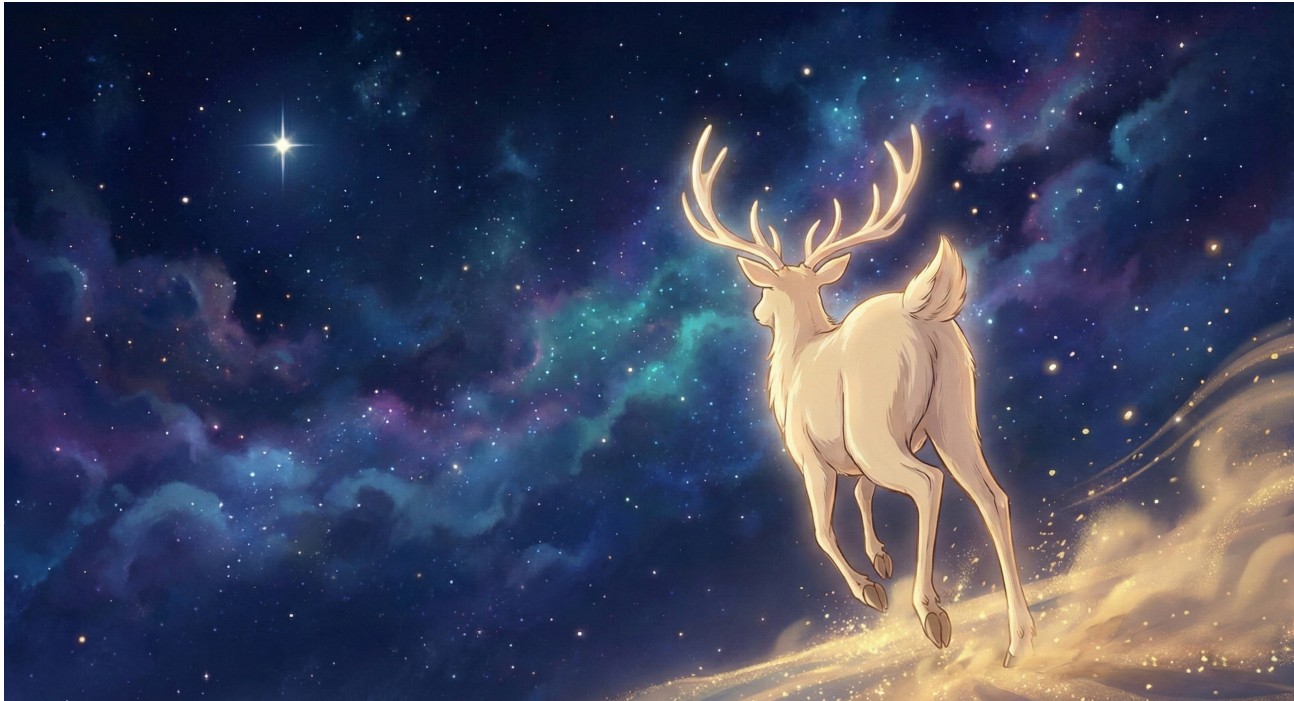

*Figure 4.* MOOSE-Star Concept Figure

The name MOOSE-Star reflects the intuition behind our framework: discovering the right inspiration from a massive scientific corpus is like finding a specific star in a vast universe. The challenge is not only the size of the search space, but also how to navigate it efficiently.

## B. Theoretical Derivation: Rigorous Decomposition of $P(h \mid b)$ via Hierarchical MDP

### B.1. Extended Fundamental Assumption

Building upon the foundational assumption in (Yang et al., 2025b) that scientific hypotheses originate from background knowledge and inspirations, we introduce the concept of *Motivation* ($m$). We posit that the discovery process is not merely a selection of entities, but a hierarchical decision process: before identifying a specific inspiration $i$ (e.g., a specific paper or entity), a researcher first formulates a high-level strategic intent or motivation $m$ (e.g., a methodology category or research direction).

We translate this extended assumption into the following functional form:

$$h = f(b, (m_1, i_1), \ldots, (m_k, i_k)) \tag{10}$$

Here, $m_j$ serves as the semantic anchor or abstract strategy for the $j$-th step, and $i_j$ is the concrete inspiration instantiation conditioned on $m_j$. The tuple $(m_j, i_j)$ represents a composite unit of inspiration for each step.

Consistent with the previous theoretical framework (Yang et al., 2025b), we maintain two critical assumptions to ensure mathematical tractability by collapsing the otherwise intractable marginalization space:

1. **Uniqueness Assumption:** Each valid hypothesis $h$ corresponds to a unique minimal set of motivation-inspiration pairs, $\{(m_1, i_1), \ldots, (m_k, i_k)\}$. *Motivation:* Instead of assuming a hypothesis is an arbitrary, intractable amalgamation of global knowledge, this posits that a discovery is driven by a specific, identifiable set of core insights. This allows us to concentrate the probability mass entirely on this single optimal set of ingredients.

2. **Fixed-Order Assumption:** The integration of this set follows a canonical constructive sequence (e.g., chronologically or by logical dependency). *Motivation:* In empirical sciences, the order in which inspirations are integrated can often be interchangeable. However, adopting a fixed canonical order serves to simplify the mathematical exposition by eliminating the need to marginalize over all $k!$ possible permutations of the selected pairs, effectively reducing the generation process to a single, deterministic reasoning path. (We provide the generalized formulation relaxing this assumption at the end of this section).

## B.2. Hierarchical Markov Decision Process Formulation

We formalize the hypothesis generation process as a *Hierarchical Markov Decision Process (Hierarchical MDP)*. Unlike a standard MDP where the action is a single step, we define the decision at step $j$ as a hierarchical process involving a high-level planner and a low-level executor:

- **State:** $s_{j-1} = (b, h_{j-1})$, representing the background context and the intermediate hypothesis generated so far (with $h_0 = \emptyset$).

- **High-level Action (Motivation):** $m_j \in \mathcal{M}$, selected by a high-level policy $\pi_{high}$. This action abstracts the search space.

- **Low-level Action (Inspiration):** $i_j \in \mathcal{I}$, selected by a low-level policy $\pi_{low}$. This action is conditioned on $m_j$.

- **Transition (Composition):** The environment updates the state to $s_j = (b, h_j)$ by integrating the selected inspiration $i_j$ into the previous intermediate hypothesis $h_{j-1}$ under the guidance of the motivation $m_j$.

## B.3. Derivation of the Decomposition

Our goal is to decompose the intractable marginal likelihood $P(h \mid b)$. Let $I$ denote the deterministic global knowledge base. Since it serves as a fixed, omnipresent context for the discovery process, conditioning on the background $b$ is practically equivalent to conditioning on the joint context $(b, I)$.

Under the *uniqueness* and *fixed-order* assumptions, the marginalization over all possible latent variables collapses to the joint probability of the unique constructive path defined by the sequence $\{(m_1, i_1), \ldots, (m_k, i_k)\}$:

$$P(h \mid b) = P(m_1, i_1, \ldots, m_k, i_k, h_1, \ldots, h_k \mid b) \tag{11}$$

$$= \prod_{j=1}^{k} P(m_j, i_j, h_j \mid b, m_1, i_1, h_1, \ldots, m_{j-1}, i_{j-1}, h_{j-1}) \tag{12}$$

$$\approx \prod_{j=1}^{k} P(m_j, i_j, h_j \mid b, h_{j-1}, I) \tag{13}$$

Equation 11 establishes the joint distribution of the unique path. Equation 12 applies the standard chain rule over the time steps $j = 1 \ldots k$, explicitly capturing the full historical trajectory up to step $j - 1$. Equation 13 applies the *Markov property*, assuming the decision at step $j$ depends primarily on the research background $b$, the global knowledge base $I$ (introduced here as the explicit search space), and the immediate previous hypothesis state $h_{j-1}$.

**Intra-step Decomposition.** Now, we perform the decomposition for the joint term at a single step $j$, $P(m_j, i_j, h_j \mid b, h_{j-1}, I)$. We first apply the exact chain rule and then apply conditional independence assumptions inherent to the scientific discovery process:

$$P(m_j, i_j, h_j \mid b, h_{j-1}, I)$$
$$= P(m_j \mid b, h_{j-1}, I) \cdot P(i_j \mid b, h_{j-1}, m_j, I) \cdot P(h_j \mid b, h_{j-1}, m_j, i_j, I) \tag{14}$$
$$\approx \underbrace{P(m_j \mid b, h_{j-1})}_{\text{Step 1: Planning}} \cdot \underbrace{P(i_j \mid b, h_{j-1}, m_j, I)}_{\text{Step 2: Retrieval}} \cdot \underbrace{P(h_j \mid b, h_{j-1}, m_j, i_j)}_{\text{Step 3: Composition}} \tag{15}$$

The transition from Equation 14 to 15 is justified by two independence properties:

1. **Internal Planning Independence:** The strategic motivation $m_j$ is an internal cognitive state generated by the model's reasoning over the context $(b, h_{j-1})$. It is conditionally independent of the external database $I$ given the context $(P(m \mid \dots, I) \approx P(m \mid \dots))$.

2. **Local Composition Independence:** Once a specific inspiration $i_j$ is retrieved, the formation of the next hypothesis state $h_j$ relies on the semantic content of $i_j$ and the current context. The rest of the global database $I$ becomes redundant $(P(h \mid \dots, i_j, I) \approx P(h \mid \dots, i_j))$.

Substituting Equation 15 back into Equation 13, we obtain the final tractable objective:

$$P(h \mid b) \approx \prod_{j=1}^{k} \left[ \underbrace{P(m_j \mid b, h_{j-1})}_{\text{Motivation Planning}} \cdot \underbrace{P(i_j \mid b, h_{j-1}, m_j, I)}_{\text{Inspiration Retrieval}} \cdot \underbrace{P(h_j \mid b, h_{j-1}, m_j, i_j)}_{\text{Hypothesis Composition}} \right] \tag{16}$$

**Generalization without the Fixed-Order Assumption.** While the fixed-order assumption significantly simplifies the theoretical exposition, in practice, the components of a scientific hypothesis can often be assembled in various valid sequences. Relaxing this assumption requires marginalizing over all valid permutations $\Pi_k$ of the $k$ steps. The generalized decomposition for MOOSE-Star is thus:

$$P(h \mid b) = \sum_{\pi \in \Pi_k} P(m_{\pi(1)}, i_{\pi(1)}, \dots, m_{\pi(k)}, i_{\pi(k)}, h_1^{(\pi)}, \dots, h_k^{(\pi)} \mid b)$$

$$\approx \sum_{\pi \in \Pi_k} \prod_{j=1}^{k} \left[ P(m_{\pi(j)} \mid b, h_{j-1}^{(\pi)}) \cdot P(i_{\pi(j)} \mid b, h_{j-1}^{(\pi)}, m_{\pi(j)}, I) \cdot P(h_j^{(\pi)} \mid b, h_{j-1}^{(\pi)}, m_{\pi(j)}, i_{\pi(j)}) \right] \tag{17}$$

where $h_0^{(\pi)} = \emptyset$, $h_k^{(\pi)} = h$, and $h_j^{(\pi)}$ denotes the intermediate hypothesis state at step $j$ under the specific permutation $\pi$.

## C. Ablation of Inspiration Retrieval Experiment: Genuine Retrieval vs. Distribution Shortcuts

A potential concern with the inspiration retrieval (IR) evaluation (Table 1) is that the model may exploit distributional patterns in the candidate pool rather than performing genuine content-based retrieval. In the standard setting, each sample contains 1 ground-truth inspiration and 14 negatives, where $\sim$35% of negatives are *hard* (keyword-overlap or embedding-similar to the ground truth) and $\sim$65% are randomly sampled. Since training and test sets use the same negative construction pipeline, the model could potentially learn a shortcut: detecting the cluster of semantically related papers and selecting from it, without truly understanding the research context.

To investigate this, we construct three ablation test sets by modifying only the negative candidates while keeping the ground truth and query context identical:

- **All Random**: All 14 negatives are randomly sampled (no hard negatives). Without a cluster of related papers in the pool, a model that relied on cluster detection loses its signal, while a model doing genuine content matching benefits from the easier distractors.

- **All Hard**: All 14 negatives are hard negatives (keyword/embedding similar to the ground truth). This is the most challenging setting, removing easy negatives that can be trivially eliminated.

- **Decoy Cluster**: In the original setting, the hard negatives are semantically related to the ground-truth inspiration, forming a cluster of topically similar papers around the correct answer. To test whether models simply detect this cluster, we replace these hard negatives with those harvested for a *different* test sample's ground-truth inspiration (the "decoy"). This produces a cluster of papers that are semantically coherent with each other—but irrelevant to the current query's research context and correct answer. The remaining slots are filled with random papers, preserving the same hard/random ratio ($\sim$35/65%) as the original. A model that selects by detecting "which group of papers looks related to each other" will be drawn to the decoy cluster; a model that matches candidates against the query context will ignore it.

Table 6 shows the results. The "Original" column is the standard setting from Table 1. Three consistent patterns emerge:

**(1) All Random $\gg$ Original**: All models show substantially higher accuracy (e.g., MS-IR-7B: 54.37% $\rightarrow$ 78.58%), confirming that hard negatives are the primary source of task difficulty.

**(2) All Hard $\approx$ Original**: Accuracy remains similar (e.g., MS-IR-7B: 54.37% vs. 53.42%), indicating that the original $\sim$35% hard negative ratio already captures most of the difficulty; adding more hard negatives has diminishing effect.

**(3) Decoy Cluster $\approx$ All Random**: Accuracies under Decoy Cluster closely match All Random across all models, demonstrating that retrieval decisions are based on content matching between the query context and candidate papers, not on detecting clusters of semantically related candidates.

These results confirm that MS-IR-7B's strong performance on the original benchmark reflects genuine inspiration retrieval ability acquired through SFT training, not exploitation of distributional artifacts in the negative construction. MS-7B exhibits the same pattern, confirming that multi-task training preserves this genuine retrieval ability.

*Table 6.* IR ablation study: accuracy (%) under modified negative construction.

| Model | Original | All Random | All Hard | Decoy Cluster |
|---|---|---|---|---|
| *Frontier Models* | | | | |
| GPT-4O-MINI | 41.99 | 77.43 | 41.00 | 77.43 |
| CLAUDE-SONNET-4.6 | 45.02 | **91.44** | **58.69** | **92.38** |
| DEEPSEEK-R1 | 45.11 | 75.29 | 43.84 | 75.64 |
| GPT-4O | 48.37 | 82.18 | 49.43 | 82.02 |
| GEMINI-3-FLASH | 51.44 | 78.86 | 48.05 | 80.84 |
| GPT-5.4 | 51.50 | 78.93 | 51.34 | 80.97 |
| GEMINI-3-PRO | **54.89** | 84.71 | 54.66 | 85.60 |
| *Baselines* | | | | |
| Random Selection | 6.70 | 6.70 | 6.70 | 6.70 |
| R1-DISTILLED-QWEN-7B | 28.42 | 45.27 | 25.30 | 42.71 |
| MS-IR-7B | **54.37** | **78.58** | **53.42** | **77.65** |
| MS-7B | 54.34 | 79.25 | 52.52 | 78.38 |

## D. Additional Analysis of Decomposed Sampling Rates

Table 5 shows that HC has a mild upward drift as $k$ increases, while BF collapses sharply. Here we provide additional diagnostics to clarify that this drift does not affect the main conclusion: decomposed subtasks maintain bounded per-step success rates, whereas BF must jointly satisfy all $k$ target deltas.

**Per-dimension breakdown by $k$.** This analysis and the following step-position analysis reuse the full evaluation data from Table 5; M3 scores are therefore computed by the same R1-DISTILLED-QWEN-32B judge. Table 7 reports average M3 sub-scores over *atomic target-delta comparisons*. For a paper with $k$ target deltas $\{\Delta h_1^*, \ldots, \Delta h_k^*\}$, each BF attempt generates one full hypothesis, which is separately scored against every target delta; thus one BF attempt contributes $k$ M3 score triples. In contrast, each HC attempt is generated for a specific step $j$ and is scored only against its assigned target $\Delta h_j^*$. For each row $k$, we pool all such scored comparison events from papers with $k$ inspirations and average their Motivation, Mechanism, Methodology, and Total scores. With this event-level averaging, BF scores drop sharply from $k = 1$ to multi-step cases and remain low at larger $k$. A single BF output must recover all $k$ innovations in one unguided sample, so the probability of matching every target delta compounds roughly as $p^k$. In contrast, HC remains comparatively stable because each comparison event evaluates one local composition step; its mild pass-rate drift is mainly associated with methodology rather than a broad increase across all dimensions.

**HC by step position.** The previous analysis groups HC attempts by the total number of inspirations $k$ in the source paper. We also analyze whether the trend is driven by step position by regrouping the same HC attempts by step index $j$. Here, $j = 0$ denotes the first composition step with no previous hypothesis state, while $j = 1$ and $j = 2$ denote later steps conditioned on the accumulated previous-hypothesis context $h_{j-1}$, which concatenates all earlier deltas. Table 8 shows

*Table 7.* Per-dimension M3 scores by inspiration depth $k$. BF scores drop sharply from single-step to multi-step cases, while HC remains comparatively stable.

| | **BF** | | | | **HC** | | | |
|---|---|---|---|---|---|---|---|---|
| $k$ | Mot | Mec | Met | Total | Mot | Mec | Met | Total |
| 1 | 2.05 | 0.91 | 0.90 | 3.86 | 2.48 | 1.83 | 1.47 | 5.77 |
| 2 | 1.49 | 0.74 | 0.70 | 2.93 | 2.28 | 1.72 | 1.52 | 5.52 |
| 3 | 1.27 | 0.78 | 0.66 | 2.71 | 2.25 | 1.76 | 1.55 | 5.56 |

that the mean total score remains similar across steps, while the pass rate increases moderately at later steps. The pass-rate increase is most aligned with the methodology dimension, consistent with the intended state-conditioned formulation: later HC steps can use $h_{j-1}$ as a more developed experimental context for specifying the next methodological increment.

*Table 8.* HC per-attempt scores by step index $j$ across PMIDs with $k \leq 3$. Later steps condition on $h_{j-1}$; the mild pass-rate increase is mainly associated with methodology, while total scores remain similar.

| **Step** $j$ | **n** | **Pass%** | **Mot** | **Mec** | **Met** | **Total** |
|---|---|---|---|---|---|---|
| 0 | 854,067 | 20.22% | 2.44 | 1.76 | 1.46 | 5.66 |
| 1 | 580,894 | 24.71% | 2.13 | 1.72 | 1.58 | 5.43 |
| 2 | 117,663 | 29.67% | 2.19 | 1.80 | 1.65 | 5.64 |

*Table 9.* Controlled ablation on the roles of the background survey context and the previous hypothesis state $h_{j-1}$, evaluated on the same 500 $k$=3 PMIDs. The research question is retained in all conditions. All samples in this diagnostic are regenerated by R1-DISTILLED-QWEN-32B and scored by GEMINI-3-FLASH using the Total $\geq 8/12$ threshold. For BF, each generation is scored separately against every $\Delta h_j^*$ for the per-delta average, while joint pass requires all three comparisons to pass.

| **Condition** | **n** | **Pass%** | **Mot** | **Mec** | **Met** | **Total** |
|---|---|---|---|---|---|---|
| *With background survey context* | | | | | | |
| HC step 0 baseline | 1998 | 33.7% | 2.95 | 2.00 | 1.62 | 6.57 |
| HC step 1 without $h_{j-1}$ | 1996 | 29.2% | 2.64 | 1.96 | 1.56 | 6.16 |
| HC step 1 with $h_{j-1}$ | 1998 | 37.9% | 2.68 | 2.14 | 1.79 | 6.62 |
| HC step 2 without $h_{j-1}$ | 1996 | 32.8% | 2.64 | 2.04 | 1.56 | 6.24 |
| HC step 2 with $h_{j-1}$ | 1999 | 44.2% | 2.75 | 2.22 | 1.83 | 6.80 |
| BF (per-delta avg.) | 5992 | 15.4% | 2.16 | 1.29 | 1.14 | 4.59 |
| BF joint pass (3-of-3) | 1992 | 1.71% | — | — | — | — |
| *Without background survey context (survey replaced by "Not provided.")* | | | | | | |
| HC step 0 baseline | 1996 | 22.5% | 2.53 | 1.88 | 1.48 | 5.89 |
| HC step 1 without $h_{j-1}$ | 1998 | 22.6% | 2.37 | 1.85 | 1.42 | 5.63 |
| HC step 2 without $h_{j-1}$ | 1998 | 24.6% | 2.39 | 1.93 | 1.45 | 5.77 |
| BF (per-delta avg.) | 5996 | 5.6% | 1.46 | 0.89 | 0.80 | 3.16 |
| BF joint pass (3-of-3) | 1996 | 0.45% | — | — | — | — |

**Controlled ablation diagnostic on the background survey and $h_{j-1}$.** The analyses above use the original sampling setting. To further isolate the role of contextual inputs, we run controlled ablations on the background survey and the accumulated previous-hypothesis context $h_{j-1}$ using the same 500 $k$=3 PMIDs. For each PMID under each condition, we regenerate 4 samples using R1-DISTILLED-QWEN-32B and re-score the valid outputs with GEMINI-3-FLASH as an independent judge. In all conditions, the research question is retained; the background-survey ablation replaces only the survey component with "Not provided", and the $h_{j-1}$ ablation removes the accumulated previous deltas. Pass rate is computed using the same Total $\geq 8/12$ threshold. Because this diagnostic uses a 500-PMID subset and an independent judge rather than the full Table 5 evaluation data, it is not used for the main pass-rate comparison; it is used only to interpret the source of the HC drift and the role of context.

Table 9 shows two patterns. First, under the original background survey, removing the accumulated $h_{j-1}$ context keeps later-step HC pass rates in the same range as the step-0 baseline: 29.2% at $j = 1$ and 32.8% at $j = 2$, compared with 33.7% at $j = 0$. Restoring $h_{j-1}$ raises pass rates to 37.9% and 44.2%, respectively, with the largest sub-score gains in methodology. Second, removing the background survey lowers absolute pass rates for both HC and BF, confirming that the survey provides useful directional context. However, even without the background survey, HC remains in a bounded per-step range (22.5–24.6%), whereas BF joint pass remains near zero (0.45%).

**Interpretation.**   Together, the breakdown and controlled ablations suggest that the mild HC drift reflects the state-conditioned nature of sequential composition, while the background survey provides a useful but not decisive directional prior. The previous hypothesis state $h_{j-1}$ provides context for later methodological specification: during distillation, this context is derived from the reference trajectory, while during inference, the same role is played by the model's accumulated hypothesis state from earlier steps. The background survey also raises absolute success rates for both BF and HC, indicating that the main Table 5 comparison is conducted in a context-rich setting rather than an artificially weakened BF setting. Crucially, with the research question retained and the background survey or $h_{j-1}$ context ablated, HC remains in a bounded per-step regime. By contrast, on the same $k=3$ PMIDs, BF joint pass remains near zero both with the background survey (1.71%) and without it (0.45%), consistent with the rapid depth-wise collapse observed in Table 5.

## E. A Failure-Probability Perspective on Brute-Force Saturation

Recent inference-scaling analyses provide a useful abstraction for understanding repeated sampling as a coverage process over instance-level failure probabilities (Levi, 2025). Consider $n$ instances, and suppose that each instance $i$ is sampled $T$ times by the brute-force generator. Let $p_i$ denote the probability that a single brute-force attempt fails on instance $i$. Under an independent-attempt approximation, the probability that all $T$ attempts fail on instance $i$ is $p_i^T$, so the probability that at least one attempt succeeds is $1 - p_i^T$. Averaging over instances yields

$$\text{pass@}T = \frac{1}{n} \sum_{i=1}^{n} (1 - p_i^T) = 1 - \frac{1}{n} \sum_{i=1}^{n} p_i^T.$$

This expression highlights that test-time scaling is governed not only by the average single-attempt success rate, but by the full distribution of instance-level failure probabilities. Easy instances with small $p_i$ are covered quickly, whereas hard-tail instances with $p_i \approx 1$ can remain unsolved even under large sampling budgets. Correlations among repeated samples would only strengthen this effect, since the effective number of independent attempts can be smaller than the nominal sampling budget.

This view clarifies why the non-trivial $k = 1$ BF pass rate in Table 5 is compatible with the saturation of brute-force sampling in Figure 3. Here, the $k$ in Table 5 denotes inspiration depth rather than the number of repeated attempts $T$. The table reports micro-averaged per-attempt pass rates, so the BF rate at $k = 1$ can reflect a subset of relatively easy instances for which the background alone already provides sufficient direction; however, this micro-average does not imply that the per-instance success probability is uniformly distributed across papers. In contrast, Figure 3 measures paper-level cumulative coverage under repeated sampling. Its saturation under large practical BF budgets indicates that many instances remain difficult to cover, a pattern naturally captured by hard-tail failure probabilities.

MOOSE-Star provides a different scaling mechanism from brute-force sampling. BF repeatedly samples complete hypotheses directly from $P(h \mid b)$, without explicitly exposing the latent inspirations that make the hypothesis reachable. As a result, each successful BF sample must implicitly solve two coupled problems at once: identifying the relevant inspirations and composing them into a valid hypothesis. When either part is unlikely, the per-attempt success probability can be near zero, so even a large number of samples may fail to cover the corresponding hard-tail instances.

By contrast, MOOSE-Star converts discovery from unguided hypothesis sampling into structured search over an explicit inspiration corpus, followed by conditioned composition. This changes the failure-probability profile in two ways. First, HC targets a single increment $\Delta h_j$ rather than the joint distribution of all $k$ innovations, avoiding the compounded success requirement of generating the entire hypothesis in one shot. Second, conditioning on $i_j$ delegates the global search problem to IR and turns generation into a guided local composition problem, which can substantially reduce the per-attempt failure probability of the composition step. If a useful inspiration—or a sufficiently close proxy—exists in the finite corpus, then retrieval is finite-reachable in the exhaustive limit; hierarchical search improves this process by prioritizing promising semantic regions under practical budgets. Once such an inspiration is retrieved, repeated HC attempts operate on a much

more constrained problem than BF: instead of inferring the missing direction from $b$ alone, the model only needs to use the retrieved inspiration to produce the corresponding hypothesis increment.

In the failure-probability view, these two effects reduce the effective difficulty of many instances. Cases that are near-zero-probability under end-to-end BF can become more tractable after the search variable is externalized and the composition target is localized. This provides one explanation for why moderate per-attempt HC pass rates can be consistent with the more favorable test-time trend observed for MOOSE-Star, while noting that Figure 3 uses a paper-level pairwise evaluation protocol rather than the per-attempt M3 threshold in Table 5.

Thus, Table 5 and Figure 3 should be read as complementary rather than interchangeable measurements. The former reports micro-averaged per-attempt success under an absolute M3 threshold, whereas the latter evaluates paper-level cumulative coverage under repeated sampling and pairwise comparison. Since they also differ in model setting and evaluation protocol, the absolute numbers are not directly comparable. The key structural point is that a non-trivial average BF pass rate does not imply scalable paper-level coverage. The saturation observed in Figure 3 indicates that, under practical brute-force budgets, many papers remain effectively hard-tail instances with near-zero cumulative success probability, even though BF can still produce valid samples on easier or more directly recoverable cases.

## F. Temporal Generalization on Temporally Held-Out Papers

To complement the test-time scaling analysis in § 7.3, we further examine whether MOOSE-Star can operate on genuinely temporally held-out scientific papers. Concretely, all evaluations in this section are conducted on papers published in October 2025, which are excluded from both the base model's pre-training period and our post-training data. This setting allows us to assess whether the framework can retrieve key inspirations and use them to construct high-quality hypotheses for future papers whose scientific content is unseen during both pre-training and post-training.

The goal of this section is not to claim prospective real-world validation, which requires waiting for future publications, but rather to provide controlled temporal out-of-distribution (OOD) evidence under a strict chronological split. We first present quantitative results on held-out future papers, measuring retrieval success and downstream hypothesis composition quality. We then analyze whether the main bottleneck lies in retrieval or composition. Finally, we provide a full $k > 1$ case study showing how MOOSE-Star retrieves two ground-truth inspirations from the inspiration corpus and composes a two-step hypothesis trajectory that closely aligns with the actual paper.

### F.1. Quantitative Evidence on Temporally Out-of-Distribution Data

Our test set consists entirely of papers published in October 2025. Both MS-IR-7B and MS-HC-7B are derived from R1-DISTILLED-QWEN-7B and fine-tuned exclusively on pre-October 2025 data. We therefore regard this evaluation as a strict temporal OOD setting.

We define a temporal reconstruction success using two criteria: (1) the hierarchical search ranks the ground-truth inspiration within the top-$K$ candidates, and (2) the HC module, when conditioned on the ground-truth inspiration, produces a hypothesis that reaches at least a target M3 score, as judged by GPT-4o. In this evaluation, retrieval is performed over an inspiration corpus containing 3,035 candidates. We evaluate this criterion on 200 test samples using MS-IR-7B for retrieval and several hypothesis composition backends. In addition to the single-task MS-HC-7B, we include MS-7B, a multitask variant jointly trained on both IR and HC (w/ $1\times$ bounded) data, which consistently outperforms the single-task model (e.g., $17.5\%$ vs. $15.0\%$ at Rank $\leq 50$, M3 $\geq 5$).

Table 10 reports the resulting success rates under three retrieval thresholds: Rank $\leq 25$, Rank $\leq 50$, and Rank $\leq 100$. These thresholds correspond to retrieving the correct inspiration within approximately the top $0.8\%$, $1.6\%$, and $3.3\%$ of the inspiration corpus, respectively.

Several observations emerge from these results. First, retrieval appears to be the main bottleneck. Among the 200 temporally held-out test samples, the ground-truth inspiration is ranked within the top 25 for $20.0\%$ of cases, within the top 50 for $28.0\%$, and within the top 100 for only slightly more cases ($29.0\%$).

Second, hypothesis composition quality still matters substantially once the correct inspiration has been identified. Under Rank $\leq 50$, MS-HC-7B achieves $4.0\%$ success at M3 $\geq 7$, while stronger frontier models used as HC backends achieve substantially higher rates ($13.5\%$–$21.0\%$). This gap indicates that stronger composition models can make more effective use of the scientific signal once the relevant inspiration is available.

*Table 10.* "Retrieval success" denotes the fraction of all 200 test cases whose ground-truth inspiration is retrieved within Rank $\leq K$. Other entries report the fraction of all 200 test cases for which this retrieval condition holds and the HC module, when conditioned on the ground-truth inspiration, reaches the corresponding score threshold.

| Rank | HC Score | MS-HC-7B | MS-7B | GPT-4o | GPT-5.4 | DeepSeek-R1 | Claude-Sonnet-4.6 | Gemini-3-Pro |
|------|----------|----------|-------|--------|---------|-------------|-------------------|--------------|
| $\leq 25$ | $\geq 4$ | 14.0% | 17.0% | 19.0% | 19.0% | 18.5% | 18.5% | 18.5% |
| | $\geq 5$ | 10.0% | 14.0% | 18.0% | 18.5% | 17.0% | 18.5% | 18.0% |
| | $\geq 6$ | 8.0% | 10.0% | 14.5% | 16.5% | 14.0% | 16.0% | 16.0% |
| | $\geq 7$ | 2.0% | 4.5% | 10.5% | 16.0% | 13.0% | 16.0% | 14.5% |
| | $\geq 8$ | 0.0% | 0.5% | 5.5% | 12.5% | 7.5% | 11.0% | 8.5% |
| | $\geq 9$ | 0.0% | 0.0% | 2.0% | 8.0% | 3.0% | 4.0% | 4.0% |
| | $\geq 10$ | 0.0% | 0.0% | 1.0% | 7.0% | 1.5% | 2.5% | 1.5% |
| | $\geq 11$ | 0.0% | 0.0% | 0.0% | 3.5% | 0.0% | 0.0% | 0.5% |
| | $\geq 12$ | 0.0% | 0.0% | 0.0% | 1.5% | 0.0% | 0.0% | 0.0% |
| | *Retrieval success* | | | | *20.0%* | | | |
| $\leq 50$ | $\geq 4$ | 20.0% | 22.5% | 26.0% | 25.5% | 25.0% | 25.0% | 25.0% |
| | $\geq 5$ | 15.0% | 17.5% | 23.5% | 25.0% | 23.5% | 25.0% | 24.0% |
| | $\geq 6$ | 12.5% | 13.0% | 19.0% | 22.5% | 20.5% | 22.0% | 21.5% |
| | $\geq 7$ | 4.0% | 5.0% | 13.5% | 20.5% | 18.0% | 21.0% | 19.5% |
| | $\geq 8$ | 0.5% | 0.5% | 6.5% | 16.0% | 10.5% | 13.0% | 11.5% |
| | $\geq 9$ | 0.0% | 0.0% | 2.5% | 10.5% | 5.0% | 4.5% | 5.5% |
| | $\geq 10$ | 0.0% | 0.0% | 1.5% | 8.5% | 2.0% | 3.0% | 1.5% |
| | $\geq 11$ | 0.0% | 0.0% | 0.5% | 4.0% | 0.0% | 0.5% | 0.5% |
| | $\geq 12$ | 0.0% | 0.0% | 0.0% | 2.0% | 0.0% | 0.5% | 0.0% |
| | *Retrieval success* | | | | *28.0%* | | | |
| $\leq 100$ | $\geq 4$ | 20.5% | 23.5% | 27.0% | 26.5% | 26.0% | 26.0% | 26.0% |
| | $\geq 5$ | 15.5% | 17.5% | 24.0% | 26.0% | 24.5% | 26.0% | 25.0% |
| | $\geq 6$ | 13.0% | 13.0% | 19.5% | 23.0% | 21.5% | 23.0% | 22.5% |
| | $\geq 7$ | 4.0% | 5.0% | 14.0% | 21.0% | 19.0% | 22.0% | 20.5% |
| | $\geq 8$ | 0.5% | 0.5% | 6.5% | 16.5% | 11.0% | 13.0% | 11.5% |
| | $\geq 9$ | 0.0% | 0.0% | 2.5% | 10.5% | 5.5% | 4.5% | 5.5% |
| | $\geq 10$ | 0.0% | 0.0% | 1.5% | 8.5% | 2.0% | 3.0% | 1.5% |
| | $\geq 11$ | 0.0% | 0.0% | 0.5% | 4.0% | 0.0% | 0.5% | 0.5% |
| | $\geq 12$ | 0.0% | 0.0% | 0.0% | 2.0% | 0.0% | 0.5% | 0.0% |
| | *Retrieval success* | | | | *29.0%* | | | |

### F.2. Case Study: A Two-Inspiration Discovery Reconstructed from a Pre-2025 Inspiration Corpus

We next present a case study on a temporally unseen paper requiring $k = 2$ inspirations. This example is informative because it exposes the full sequential structure of MOOSE-Star: the framework must retrieve one ground-truth inspiration, compose a meaningful intermediate hypothesis, and then retrieve and integrate a second ground-truth inspiration that extends the scientific trajectory. Unlike a $k = 1$ example, this setting more directly illustrates how multiple inspirations can be combined into a coherent multi-step discovery process.

**Target paper.** *TREM2 Impedes Recovery After Spinal Cord Injury by Regulating Microglial Lysosomal Membrane Permeabilization-Mediated Autophagy (Cell Proliferation, October 2025; DOI: 10.1111/cpr.70047).*

**Research question.**

How to enhance microglial survival and functional recovery after spinal cord injury by regulating lysosomal integrity and autophagy flux?

**Background survey.**

Spinal cord injury (SCI) involves irreversible primary mechanical damage followed by secondary injury phases. Microglia, the primary immune responders in the CNS, proliferate post-injury and their depletion exacerbates damage.

Protecting microglial numbers is a known therapeutic strategy, but mechanisms regulating their survival/function remain unclear. TREM2 regulates microglial activation states (e.g., DAM phenotype) in neurodegenerative diseases but has not been studied in SCI. Single-cell data suggest a link between TREM2 and lysosomal pathways in microglia. Lysosomal dysfunction (reduced activity, impaired autophagy) contributes to neuronal death in SCI, but its role in microglial survival is unexplored. Autophagy maintains cellular homeostasis but requires functional lysosomes.

**Retrieved inspirations.** The two retrieved ground-truth inspirations by MS-IR-7B are shown in Table 11.

*Table 11*. Retrieved ground-truth inspirations for the temporally held-out TREM2 case. For this $k = 2$ example, MOOSE-Star ranks the two ground-truth inspirations at 12/3035 and 1/3035, respectively.

| Step | Ground-Truth Inspiration | DOI | Retrieval Rank |
|---|---|---|---|
| 0 | *Underlying Mechanism of Lysosomal Membrane Permeabilization in CNS Injury: A Literature Review* (*Molecular Neurobiology*, 2024) | 10.1007/s12035-024-04290-6 | **12 / 3035** |
| 1 | *TFEB links autophagy to lysosomal biogenesis* (Settembre et al., *Science*, 2011) | 10.1126/science.1204592 | **1 / 3035** |

**Step 0: Generated Hypothesis (GPT-5.4 as HC model; M3 = 9/12).**

Inspiration: Lysosomal membrane permeabilization (LMP) is the key upstream lysosomal lesion in SCI microglia, and stabilizing lysosomal membranes to prevent cathepsin leakage will preserve autophagy flux and microglial survival.

- Motivation (WHY): Current SCI approaches describe lysosomal dysfunction and impaired autophagy broadly, but they do not specify the proximal lysosomal failure that converts injury stress into microglial death and loss of function. The inspiration paper identifies LMP as a central event in CNS injury that links oxidative/inflammatory stress to release of lysosomal hydrolases, programmed cell death, and autophagy disruption. Adapting this to SCI suggests that microglial loss may be driven not simply by reduced lysosome activity, but by physical destabilization of lysosomal membranes. Targeting LMP therefore fills the mechanistic gap between SCI-induced stress and both outcomes of interest: diminished microglial survival and defective autophagic clearance.

- Mechanism (HOW IT WORKS): After SCI, microglia are exposed to ROS, lipid peroxidation, excitotoxic/calcium stress, and inflammatory mediators that can permeabilize lysosomal membranes. LMP causes partial or extensive escape of cathepsins and other lysosomal contents into the cytosol, which then triggers lysosome-dependent cell death and amplifies apoptosis/pyroptotic-inflammatory signaling. At the same time, membrane-compromised lysosomes lose degradative competence and acidity, so autophagosomes can no longer be efficiently processed to completion, producing stalled autophagy flux rather than protective recycling. Therefore, preserving lysosomal membrane integrity should simultaneously: (i) retain cathepsins within lysosomes, (ii) maintain lysosomal acidification and enzyme compartmentalization, (iii) allow autophagosome-lysosome degradation to proceed, and (iv) reduce microglial death and dysfunctional inflammatory activation. The net effect should be better preservation of viable, functional microglia after SCI.

- Methodology (HOW IT'S INTEGRATED): In SCI microglia, explicitly model and measure LMP as the central lysosomal integrity variable rather than only total lysosome abundance or generic autophagy markers. Induce SCI or SCI-like stress in vivo/in vitro, then quantify LMP using lysosomal rupture/permeabilization readouts such as galectin puncta recruitment, cytosolic cathepsin B/D/L redistribution, lysosomal pH loss, acridine orange or LysoTracker leakage, and ultrastructural membrane damage. In parallel, measure autophagy flux with LC3-II turnover and p62/SQSTM1 clearance under lysosomal blockade controls, and relate these to microglial survival and phenotype. Test whether lysosome-preserving interventions—such as lysosomal membrane stabilizers, cathepsin release/activity suppression, or anti-oxidative strategies that specifically reduce membrane lipid damage—rescue microglia by lowering LMP. Mechanistic validation should require showing that interventions improve survival and functional recovery only when they reduce LMP, retain lysosomal cathepsins, and restore degradative autophagy flux. This establishes LMP prevention as the actionable lysosomal-integrity strategy for enhancing microglial survival after SCI.

**Step 0: Ground-Truth Hypothesis.**

Inspiration: Lysosomal membrane permeabilization (LMP) causing cell death

- Motivation (WHY): The specific limitation addressed is the lack of understanding of how lysosomal dysfunction contributes to microglial death after spinal cord injury (SCI). Existing methods focus on protecting microglial numbers but fail to address the molecular mechanisms of microglial survival post-injury. LMP, a process where lysosomal membranes lose integrity, has been implicated in cell death in non-SCI contexts (e.g., cancer, neurodegeneration), but its role in microglial death after SCI remains unexplored. This gap is critical because lysosomal integrity is essential for maintaining autophagy, a key survival mechanism for stressed cells.

- Mechanism (HOW IT WORKS): LMP occurs when lysosomal membranes lose their integrity, leading to the leakage of lysosomal enzymes (e.g., cathepsins) into the cytoplasm. This leakage triggers a cascade of events, including the activation of pro-apoptotic proteins (e.g., galectin-3, Gal3) and the loss of lysosomal function, ultimately resulting in cell death. In the context of SCI, this mechanism is particularly relevant because microglia are exposed to severe oxidative stress and inflammatory mediators, which can compromise lysosomal membranes. The causal chain is as follows: (1) SCI induces oxidative stress and inflammation. (2) Oxidative stress disrupts lysosomal membrane integrity. (3) LMP leads to cathepsin leakage and Gal3 activation. (4) Gal3 activation triggers pro-apoptotic signaling pathways. (5) Microglial death exacerbates secondary injury.

- Methodology (HOW IT'S INTEGRATED): To investigate LMP in microglia after SCI, the following steps are implemented: (1) Experimental design: Wild-type (Wt) and $Trem2^{-/-}$ mice are subjected to spinal cord injury (SCI) using a 70 kD impact force. Microglia are isolated from the spinal cord tissue 7 days post-injury. (2) LMP induction: N-dodecylimidazole (NDI) is used as a lysosomal membrane permeabilization inducer at a concentration of 40 $\mu$M in vitro. (3) Marker assessment: Immunofluorescence staining is performed to measure the number of Gal3 puncta (a marker of lysosomal damage) and the colocalization of lysosomal markers (e.g., LAMP1) with cathepsin L (CTSL). Western blotting is used to quantify the levels of CTSL, CTSD, and CTSC in lysosomal and cytosolic fractions. (4) Quantification: The mean fluorescence intensity of Gal3 puncta is measured using ImageJ software. The ratio of lysosomal to cytosolic proteases is calculated to assess LMP severity.

**Step 1: Generated Hypothesis (GPT-5.4 as HC model; M3 = 12/12).**

Inspiration: TFEB-driven coupling of lysosomal biogenesis to autophagy flux restoration

- Motivation (WHY): The previous hypothesis identifies lysosomal membrane damage as a cause of microglial death after spinal cord injury (SCI), but it does not explain how surviving microglia could rebuild degradative capacity once lysosomes are injured or insufficient. A key remaining limitation is that lysosomal integrity and autophagy flux are still treated mainly as damage readouts rather than as a coordinated repair program. The inspiration paper suggests that TFEB is a master regulator that simultaneously induces lysosomal biogenesis and the full autophagy machinery. This is directly relevant because microglia in SCI must clear damaged organelles, myelin debris, and inflammatory cargo under intense stress; without a transcriptional program that replenishes lysosomes and reopens autophagic flux, microglia would remain vulnerable to persistent lysosomal failure and cell death. Thus, the specific contribution of this inspiration is the hypothesis that impaired TFEB activation is an upstream reason why SCI microglia fail to restore lysosomal competence, and that enhancing TFEB activity can rescue microglial survival and function.

- Mechanism (HOW IT WORKS): After SCI, microglia experience oxidative, inflammatory, and metabolic stress that increases degradative demand while damaging lysosomes. In this setting, TFEB functions as a transcriptional switch that can restore the lysosomal-autophagy system if it translocates to the nucleus. Once activated, TFEB is expected to induce a coordinated gene program covering: (1) lysosomal biogenesis and acidification (e.g., LAMP1, lysosomal hydrolases, v-ATPase-related genes), (2) autophagosome formation machinery, (3) autophagosome-lysosome fusion components, and (4) substrate degradation capacity. In SCI microglia, defective TFEB nuclear localization or persistent inhibitory phosphorylation would prevent this adaptive program, producing too few functional lysosomes, incomplete autophagic degradation, p62/LC3 cargo accumulation, and reduced capacity to clear damaged material. This would amplify lysosomal stress and sensitize cells to the lysosomal injury pathway defined in the previous hypothesis. In contrast, TFEB activation should replenish the lysosome pool, improve lysosomal enzyme content and acidification, increase fusion of autophagosomes with lysosomes, and re-establish

autophagy flux. Functionally, this should enhance microglial survival, preserve phagocytic clearance, and support a reparative response after SCI. Because TREM2 is already linked by background data to lysosomal pathways, an additional testable extension is that TREM2 may promote microglial resilience partly by permitting TFEB activation/nuclear translocation; therefore, TFEB dysfunction may be more severe in Trem2-deficient microglia.

- Methodology (HOW IT'S INTEGRATED): To integrate this concept into the existing SCI-microglia framework, TFEB is introduced as an upstream regulatory node controlling lysosomal recovery and autophagy completion: (1) Determine whether TFEB activation is impaired after SCI and in Trem2 deficiency—use wild-type and $Trem2^{-/-}$ SCI mice, measure TFEB subcellular localization by immunofluorescence and nuclear/cytoplasmic fractionation, quantify TFEB phosphorylation status, and perform qPCR/RNA-seq for TFEB target genes. (2) Assess whether TFEB status predicts restoration of lysosomal function—measure lysosome abundance using LAMP1/LAMP2 staining, lysosomal acidity using LysoTracker/LysoSensor, and compare between WT and $Trem2^{-/-}$ microglia. (3) Directly test whether TFEB restores autophagy flux—evaluate LC3-II and p62/SQSTM1 with and without lysosomal blockade (e.g., bafilomycin A1), use tandem fluorescent LC3 reporter (mCherry-GFP-LC3) to quantify autophagosome maturation. (4) Manipulate TFEB causally—activate TFEB genetically or pharmacologically, inhibit via siRNA/shRNA or CRISPR interference, test rescue of lysosomal biogenesis, autophagy flux, and cell survival, especially in $Trem2^{-/-}$ microglia. (5) Link TFEB-mediated restoration to microglial functional recovery—measure viability, phagocytosis, and inflammatory gene expression; in vivo, test whether microglia-targeted TFEB activation improves tissue sparing and behavioral recovery. (6) Integrate with the prior LMP framework—use lysosomal damage assays as downstream comparators to show that TFEB activation reduces susceptibility to lysosomal failure by increasing lysosome renewal and autophagy completion.

**Step 1: Ground-Truth Hypothesis.**

Inspiration: TFEB-mediated lysosomal biogenesis under cellular stress

- Motivation (WHY): The specific limitation addressed is the lack of understanding of how to enhance autophagy in microglia after SCI. While autophagy is recognized as a critical survival mechanism, its regulation in the context of lysosomal dysfunction remains unclear. Existing methods do not address how to activate lysosomal biogenesis in stressed microglia, which is essential for maintaining autophagic flux and preventing cell death.

- Mechanism (HOW IT WORKS): Transcription factor EB (TFEB) is a master regulator of lysosomal biogenesis and autophagy. Under cellular stress, TFEB translocates from the cytoplasm to the nucleus, where it activates the transcription of genes encoding lysosomal enzymes (e.g., CTSD) and autophagy-related proteins (e.g., LC3). This mechanism ensures that cells can degrade damaged organelles and recycle nutrients, promoting survival under stress conditions. In the context of SCI, this mechanism is particularly relevant because: (1) Microglia are exposed to oxidative stress and inflammation post-injury. (2) Oxidative stress disrupts lysosomal integrity, impairing autophagy. (3) TFEB activation can counteract this by promoting lysosomal biogenesis and restoring autophagy.

- Methodology (HOW IT'S INTEGRATED): To investigate TFEB-mediated lysosomal biogenesis in microglia after SCI: (1) Experimental design: Wild-type (Wt) and $Trem2^{-/-}$ mice are subjected to SCI. Microglia are isolated from the spinal cord tissue 7 days post-injury. (2) TFEB activation: In vitro experiments are performed using BV2 cells treated with L-Leucyl-L-Leucine methyl ester (LLOMe) to induce LMP. Small interfering RNA (siRNA) targeting TREM2 is used to silence its expression. (3) Marker assessment: Immunofluorescence staining is performed to measure TFEB nuclear translocation. Western blotting is used to quantify the levels of TFEB, p-Syk, and autophagy markers (e.g., LC3-II, P62) in nuclear and cytosolic fractions. (4) Quantification: The mean fluorescence intensity of nuclear TFEB is measured using ImageJ software. The ratio of LC3-II to P62 is calculated to assess autophagic flux.

Key Integration Between Inspirations: The two inspirations are integrated through the Syk signaling pathway. Specifically: (1) TREM2 knockout reduces Syk phosphorylation, which inhibits Syk-mediated LMP induction. (2) Reduced Syk phosphorylation promotes TFEB nuclear translocation, activating lysosomal biogenesis and autophagy. (3) This dual mechanism (LMP repair + autophagy enhancement) rescues microglial survival and promotes functional recovery after SCI.

**Discussion.** This case illustrates the full sequential structure of MOOSE-Star on a temporally unseen paper. The model input (research question plus background survey) contains no mention of LMP, TFEB, or lysosomal biogenesis; these

key scientific concepts enter the reasoning process only through the retrieved inspirations. Moreover, the two retrieved inspirations do not function as isolated fragments. Step 0 identifies LMP as the proximal mechanism of lysosomal failure, while Step 1 introduces TFEB as a regulatory program for restoring lysosomal function and autophagy flux. Together, the two steps closely reconstruct the two-part hypothesis structure of the actual paper.

## G. Limitations

This work focuses on making the hypothesis proposal distribution $P(h \mid b)$ tractably trainable through decomposition. It does not study feedback-driven refinement or experiment-guided ranking of generated hypotheses, which has been explored in complementary settings such as MOOSE-Chem3 (Liu et al., 2025a). More broadly, executable or automated feedback has also been used in mathematical, algorithmic, and equation discovery settings (Romera-Paredes et al., 2024; Novikov et al., 2025; Shojaee et al., 2025). These settings operate after candidate hypotheses have already been proposed: external feedback, such as experimental measurements, executable evaluations, or reward scores, is used to score or compare candidates and then guide selection or iterative refinement. In contrast, MOOSE-Star targets the preceding stage: making the hypothesis proposal distribution $P(h \mid b)$ itself tractably trainable before such feedback is available.

Our experiments evaluate whether the model can recover discovery-relevant inspirations and reconstruct the corresponding hypotheses in temporally held-out literature-derived cases. Prospective experimental testing of newly proposed hypotheses would require domain-specific validation campaigns, which are not included in the present study.

## H. The *M3 Rubric* for Comparing Generated Hypothesis with Groundtruth Hypothesis

**Scoring Rubric (0–4 for each dimension)**

**IMPORTANT INSTRUCTIONS:**

1. Score based on **RECALL** – what percentage of GT content is correctly covered by Generated.

2. Both **MISSING** and **WRONG** content count as "not covered".

3. The examples below are **ONLY for illustration** – do **NOT** match against them.
   Score the actual GT vs Generated content, not similarity to examples.

---

**Motivation (WHY) – Does it identify the same research gap?**

**How to score:** Count what percentage of GT's key elements appear correctly in Generated.

**[EXAMPLE FOR ILLUSTRATION ONLY – do not match against this]**

**Example GT:**

> Current deep learning methods for brain tumor segmentation in MRI scans suffer from low accuracy at tumor boundaries, particularly in low-contrast glioma regions, due to insufficient modeling of boundary uncertainty

(5 key elements: domain=brain tumor/MRI, task=segmentation, problem=low boundary accuracy, context=low-contrast glioma, cause=insufficient uncertainty modeling)

- **4 (∼100%):**

  > Existing DL approaches for brain tumor MRI segmentation have poor boundary delineation in glioma cases because they fail to model boundary uncertainty

  ✓ All 5: domain + task + problem + context + cause

- **3 (∼75%):**

  > Missing: "Brain tumor segmentation methods have low accuracy at boundaries in gliomas"

✓ 4: domain + task + problem + context — × Missing: cause

Wrong: "Brain tumor segmentation in MRI has low boundary accuracy in gliomas due to limited training data"

✓ 4: domain + task + problem + context — × Wrong: cause (limited data vs uncertainty modeling)

- **2 (∼50%):**

    Missing: "Brain tumor segmentation in MRI has accuracy issues"

✓ 2.5: domain + task + vague problem — × Missing: context + cause

Wrong: "Brain tumor detection in MRI suffers from false positives in gliomas"

✓ 2: domain + context — × Wrong: task (detection) + problem (false positives)

- **1 (∼25%):**

    Missing: "Medical image segmentation needs improvement"

✓ 1: broad domain only — × Missing: specific target + problem + context + cause

Wrong: "Brain tumor classification methods are too slow"

✓ 1: target organ — × Wrong: task (classification) + problem (speed)

- **0 (∼0%):**

    "Protein structure prediction lacks accuracy"

✓ 0 — × Completely unrelated domain

    "Natural language models struggle with long-range dependencies"

✓ 0 — × Completely unrelated domain

**Mechanism (HOW IT WORKS) – Does it propose the same core mechanism?**

**GT:**

Apply transformer-based attention with boundary-aware loss functions to learn multi-scale feature representations, enabling precise tumor boundary localization through uncertainty-guided refinement

(5 key elements: architecture=transformer attention, loss=boundary-aware, features=multi-scale, task=boundary localization, technique=uncertainty refinement)

- **4 (∼100%):**

    Use transformer attention with boundary loss for multi-scale features to localize tumor boundaries via uncertainty-guided refinement

✓ All 5: architecture + loss + features + task + technique

- **3 (∼75%):**

    Missing: "Transformer attention with boundary-aware loss for multi-scale feature learning to localize tumor boundaries"

✓ 4: architecture + loss + features + task — × Missing: refinement technique

Wrong: "Transformer attention with boundary loss for multi-scale features to localize boundaries via post-processing CRF"

✓ 4: architecture + loss + features + task — × Wrong: technique (CRF vs uncertainty)

- **2 (∼50%):**

  Missing: "Use attention mechanism with boundary loss for tumor boundary detection"

  ✓ 2.5: partial architecture + loss + task   —   × Missing: multi-scale + refinement

  Wrong: "Use transformer attention with standard loss for multi-scale feature learning"

  ✓ 2.5: architecture + features   —   × Wrong: loss (standard vs boundary) + missing: task + technique

- **1 (∼25%):**

  Missing: "Apply deep learning for tumor analysis"

  ✓ 0.5: broad method category   —   × Missing: specific architecture + loss + features + technique

  Wrong: "Use transformer attention for image classification"

  ✓ 1: architecture   —   × Wrong: task (classification) + missing: loss + features + technique

- **0 (∼0%):**

  "Apply LSTM for time series forecasting"

  ✓ 0   —   × Completely unrelated mechanism and task

  "Use rule-based heuristics for text classification"

  ✓ 0   —   × Completely unrelated mechanism and domain

**Methodology (HOW IT'S INTEGRATED) – Does it describe similar implementation?**

**GT:**

Train on BraTS 2021 dataset (1251 MRI cases), implement 3D U-Net with transformer encoder, use combined Dice-boundary loss, apply 5-fold cross-validation, evaluate with Dice score and 95% Hausdorff distance

(6 key details: dataset=BraTS 2021/1251, architecture=3D U-Net + transformer, loss=Dice-boundary, validation=5-fold CV, metrics=Dice + HD95)

- **4 (∼100%):**

  Train on BraTS 2021 (1251 scans), 3D U-Net with transformer encoder, Dice-boundary loss, 5-fold CV, report Dice and HD95

  ✓ All 6: dataset + architecture + loss + validation + metrics

- **3 (∼75%):**

  Missing: "BraTS 2021 dataset, 3D U-Net + transformer, Dice-boundary loss, 5-fold CV, Dice score"

  ✓ 5: dataset + architecture + loss + validation + partial metrics   —   × Missing: HD95

  Wrong: "BraTS 2021 (1251 scans), 3D U-Net + transformer, Dice-boundary loss, 3-fold CV, Dice and HD95"

  ✓ 5: dataset + architecture + loss + metrics   —   × Wrong: validation (3-fold vs 5-fold)

- **2 (∼50%):**

  Missing: "Train on BraTS dataset with 3D U-Net, evaluate Dice score"

  ✓ 3: partial dataset + architecture + partial metrics   —   × Missing: size + loss + validation + HD95

  Wrong: "Train on private dataset, use 2D U-Net, Dice-boundary loss, 5-fold CV, Dice"

✓ 3: loss + validation + partial metrics   —   × Wrong: dataset + architecture (2D vs 3D)

- **1 ($\sim$25%):**

   Missing: "Train a segmentation model on brain MRI data"

✓ 1.5: vague dataset + vague approach   —   × Missing: specific details

   Wrong: "Train on TCGA genomic data, use ResNet, cross-entropy, accuracy"

✓ 0.5: general training   —   × Wrong: dataset + architecture + loss + metrics

- **0 ($\sim$0%):**

   "Fine-tune GPT on dialogue dataset with RLHF"

✓ 0   —   × Completely different domain and methodology

   "Survey 200 patients using questionnaires, analyze with chi-square test"

✓ 0   —   × Completely different methodology type

