# OpenReview forum: "MOOSE-Star: Unlocking Tractable Training for Scientific Discovery by Breaking the Complexity Barrier"
_ICML.cc/2026/Conference — ICML 2026 regular_

### Official Review · Reviewer_5wGb · 2026-02-16

**Soundness:** 3
**Presentation:** 3
**Significance:** 3
**Originality:** 3
**Overall Recommendation:** 5
**Confidence:** 3

**Summary:**

This paper proposes MOOSE-STAR, a framework to effectively approximate the probability space of scientific hypotheses given a research background and existing literature. It introduces several novel techniques to make the training of this approximation tractable, such as decomposing the core equation into sequential sub-tasks: Inspiration Retrieval (IR) and Hypothesis Composition (HC). Key innovations include a guided hierarchical search over literature and bounded composition to maintain robustness against retrieval noise. The authors also release TOMATO-STAR, a substantial dataset of processed papers. The theoretical analysis demonstrating the exponential intractability of direct hypothesis discovery is a strong foundation for the work.

**Compliance With Llm Reviewing Policy:**

Affirmed.

**Final Justification:**

I have decided to increase my score to Accept because the authors successfully validated their automated evaluation pipeline with expert human judgment (even if just with a single human expert). Together with the rest of contributions, I think it is a strong paper.

**Key Questions For Authors:**

- Experimental Feedback: While MOOSE-STAR significantly improves hypothesis generation quality, feedback-based refinement is essential to the human research process. Do the authors see a way to incorporate experimental results or feedback loops into this training framework, or do they view it as strictly complementary to existing feedback-driven works?

- Dataset Quality Assurance: Regarding the TOMATO-STAR dataset, the paper mentions three automated checks (information disjointness, information sufficiency,  non-redundancy). Since these are complex semantic tasks, could the authors provide more detail on the specific implementation of these checks?

How does the trained model compare to frontier LLMs? While the paper already shows improved results over the baseline following the proposed method, it would be informative to see how the trained model compares with frontier models on hypothesis generation. This could provide valuable information about whether scaling their approach could lead to SOTA models for hypothesis generation.

**Limitations:**

The authors include a dedicated Impact Statement that adequately addresses both potential societal risks and the broader positive impacts of their work.

**Strengths And Weaknesses:**

Strengths:
- Clear Narrative: The paper is very well-motivated and clearly written, making complex probabilistic decompositions easy to follow.
- High Interest: The topic is of significant relevance to the AI for Science community, especially the challenge of moving beyond simple inference to formal training of a hypothesis generation models.
- Practical Training Recipe: this work offers a concrete, scalable way to train a hypothesis generation LLM by breaking the complexity barrier, which is a novel and valuable result. The component of the method are well grounded and tested.
- Dataset Contribution: The release of the large-scale TOMATO-STAR dataset is a relevant contribution to the field.

Weaknesses:
- Originality of Components: While the combination is novel, the core decomposition theory builds heavily on prior work (MOOSE-Chem).
- Reliance on LLM-based evaluation: The authors rely on Gemini-3-Flash to judge the quality of generated hypotheses. While they attempt to control for position bias and use structured rubrics, this remains an internal proxy for scientific validity. LLMs are prone to favouring stylistic and professional formatting over actual technical substance, and still have a dubious scientific taste. This risks rewarding ideas that look correct but have lower scientific quality. A small-scale validation by human experts would increase confidence in the technical plausibility and genuine novelty of the model outputs. This is my main concern and I will increase my score if the authors provide evidence that their automated metrics correlate with human expert judgment.

---

> ### Author Rebuttal · Authors · 2026-03-31
>
> We appreciate your insightful questions and helpful suggestions.
>
> **Q1**: The core decomposition theory builds on prior work MOOSE-Chem.
>
> **A1**: While MOOSE-Chem proposed the formula and used it for inference, this paper shows how to operationalize it for the training of P(h|b), thereby turning a previously theoretical decomposition into a concrete training recipe.
>
> **Q2**: The authors rely on Gemini-3-Flash to compare the quality of hypotheses generated by MS and BF baseline. A small-scale validation by human experts would help.
>
> **A2**: We conducted an expert validation by repeating the pairwise evaluation on 40 samples, with the comparisons annotated by a biology PhD expert.
>
> Among the 23 cases where both Gemini-3-Flash and the expert selected a decisive winner (MS or BF, excluding Tie), they agreed in 22 cases (95.7%). The only true winner reversal occurred when Gemini-3-Flash preferred BF while the expert preferred MS.
>
> For the remaining 17 cases involving at least one Tie, 5 were also marked as Tie by the expert, while the expert preferred BF in 7 cases and MS in 5 cases. Overall, these results suggest that our automated evaluation is well aligned with expert judgment, especially when the preference is decisive.
>
> **Q3**: Do the authors view MOOSE-Star as strictly complementary to existing feedback-driven works?
>
> **A3**: We view MOOSE-Star as largely complementary to existing feedback-driven work. Our focus is on strengthening the discovery **prior** P(h|b), i.e., improving hypothesis generation **before** seeing external evidence. By contrast, feedback-driven methods focus on refining or ranking hypotheses **after** observing feedback. Thus, the two directions address different stages of scientific discovery and are naturally complementary.
>
> **Q4**: Could the authors provide more detail on the specific implementation of the three automated checks?
>
> **A4**: Due to space constraints, we summarize the input and core logic of each check below.
>
> **Check 1: Information Disjointness.** Input: research question, background survey, hypothesis, and all inspirations. The LLM checks whether the background (research question and background survey) already mentions or hints at the key external concepts that constitute the inspirations.
>
> **Check 2: Information Sufficiency.** Input: research question, background survey, hypothesis, and inspiration papers (title + abstract). The LLM evaluates whether someone with the background and these inspiration papers could conceive the structural skeleton of the hypothesis.
>
> **Check 3: Non-Redundancy.** Input: research question, background survey, hypothesis, and all inspiration papers. The LLM checks whether multiple inspirations cover the same core concept, and if so, identifies which to prune.
>
> **Check 4: Information Necessity.** Input: research question, background survey, the next-step hypothesis, and the specific inspiration being checked. The LLM verifies that each inspiration is a necessary conceptual bridge between background and hypothesis — not trivial or already known.
>
> **Q5**: It would be informative to see how the trained model compares with frontier models.
>
> **A5**: We additionally compare our trained models against several frontier models on both **inspiration retrieval (IR)** and **hypothesis composition (HC)** tasks.
>
> For HC, the original results in Table 2 used R1-DISTILLED-QWEN-32B as the judge. To standardize the evaluation, we re-evaluate HC using GPT-4o as the judge, and benchmark frontier models under the same setup.
>
> **IR results.**
>
> | Model | Accuracy |
> |---|---:|
> | *Frontier Models* |  |
> | GPT-4o-mini | 41.99% |
> | Claude-Sonnet-4.6 | 45.02% |
> | DeepSeek-R1 | 45.11% |
> | GPT-4o | 48.37% |
> | Gemini-3-Flash | 51.44% |
> | GPT-5.4 | 51.50% |
> | Gemini-3-Pro | **54.89%** |
> | *Baselines* |  |
> | Random Selection | 6.70% |
> | R1-Distilled-Qwen-7B | 28.42% |
> | **MS-IR-7B** | **54.37%** |
>
> **HC results.**
>
> | Model | Total | Mot | Mec | Met |
> |---|---:|---:|---:|---:|
> | *Frontier Models* |  |  |  |  |
> | GPT-4o-mini | 5.18 | 2.26 | 1.64 | 1.28 |
> | GPT-4o | 5.99 | 2.43 | 1.95 | 1.60 |
> | DeepSeek-R1 | 6.42 | 2.54 | 2.10 | 1.77 |
> | Gemini-3-Flash | 6.42 | 2.48 | 2.16 | 1.79 |
> | Gemini-3-Pro | 6.70 | 2.59 | 2.26 | 1.85 |
> | Claude-Sonnet-4.6 | 7.28 | 2.78 | 2.41 | 2.08 |
> | GPT-5.4 | **7.82** | **2.99** | **2.61** | **2.22** |
> | *Baselines* |  |  |  |  |
> | R1-Distilled-Qwen-7B | 4.05 | 1.96 | 1.30 | 0.80 |
> | MS-HC-7B | 4.68 | 2.13 | 1.46 | 1.09 |
> | w/ 1× bounded | **4.74** | **2.16** | **1.48** | **1.10** |
> | w/ 2× bounded | 4.73 | 2.15 | **1.48** | 1.09 |
>
> Overall, we find that MS-IR-7B already reaches near-frontier performance on IR, while MS-HC-7B, although substantially stronger than the R1-Distilled-Qwen-7B baseline, still remains clearly below frontier models on HC.
>
> This suggests that knowledge association for IR may be more readily activated through post-training, whereas HC requires deeper scientific understanding and stronger base-model capacity.

---

> > ### Author Rebuttal · Reviewer_5wGb · 2026-04-01
> >
> > Thank you for your responses and the additional results. While I would have preferred to see evaluation correlations based on more than a single human expert, I agree that even one expert provides some validation, making the paper stronger overall.

---

### Official Review · Reviewer_7pxv · 2026-03-13

**Soundness:** 3
**Presentation:** 3
**Significance:** 3
**Originality:** 3
**Overall Recommendation:** 4
**Confidence:** 3

**Summary:**

This paper tackles a fundamental problem in training LLMs for scientific discovery: directly modeling P(hypothesis|background) is mathematically intractable due to combinatorial complexity O(N^k) when retrieving inspirations from large knowledge bases. The authors propose MOOSE-STAR, a framework that makes training tractable. Experiments show MOOSE-STAR scales better at test time than brute-force sampling.

**Compliance With Llm Reviewing Policy:**

Affirmed.

**Key Questions For Authors:**

see Weaknesses

**Limitations:**

yes

**Strengths And Weaknesses:**

Strengths:
1 The complexity analysis (O(N^k) → O(log N)) is clear and well-motivated. The decomposition into retrieval + composition steps is mathematically sound and practically useful.

2 Each component (hierarchical search, bounded composition, motivation planning) addresses a concrete bottleneck. The trade-off analysis is thoughtful.

3 TOMATO-STAR is a large, carefully constructed dataset with explicit quality controls.

Weaknesses
1  Most results focus on component-level metrics (IR accuracy, HC M3 scores). There's little evidence on full hypothesis generation quality or downstream scientific utility, does MOOSE-STAR actually produce better hypotheses?

2 Selecting semantic proxies via SPECTER2 similarity and prioritizing "hard" tiers for training is clever, but risks overfitting to the embedding space. How robust is bounded composition to distribution shift in retrieved candidates?

3 Training data is generated by R1-DISTILLED-QWEN-32B and filtered by a rubric-based evaluator (also LLM-based). While common, this could amplify model-specific biases.

---

> ### Author Rebuttal · Authors · 2026-03-31
>
> Thank you for your insightful inquiries. In the following sections, we've structured our responses to each of your points raised.
>
> **Q1**: Instead of component-level metrics, what about full hypothesis generation quality?
>
> **A1**: Thank you for this important question. We agree that full hypothesis generation quality is critical. In fact, we evaluate this end-to-end behavior in **Figure 4**, rather than only through component-level metrics.
>
> Specifically, the brute-force baseline directly samples hypotheses from R1-DISTILLED-QWEN-7B conditioned only on the background, without explicit inspirations. In contrast, MOOSE-Star is evaluated as an integrated framework combining **IR + HC**. Its total inference cost includes both the IR calls made during hierarchical search and the HC call made once a retrieved inspiration is used for hypothesis composition. We then compare the resulting end-to-end hypothesis quality under increasing test-time compute budgets.
>
> As shown in Figure 4, brute-force sampling quickly reaches a performance ceiling, whereas MOOSE-Star continues to improve with additional test-time compute. This is exactly an evaluation of full hypothesis generation quality, not just isolated component performance. The result suggests that structured inspiration retrieval and composition allow MOOSE-Star to use test-time compute more effectively than direct end-to-end sampling from background alone.
>
> More broadly, the key point is that MOOSE-Star converts full hypothesis generation from an unstructured sampling problem into a structured search-and-compose process. As long as the search can identify a sufficiently relevant inspiration, the downstream HC step becomes much more tractable, which is the main reason the framework scales better at the full-generation level.
>
> **Q2**: Selecting semantic proxies via SPECTER2 similarity and prioritizing "hard" tiers for training is clever, but risks overfitting to the embedding space. How robust is bounded composition to distribution shift in retrieved candidates?
>
> **A2**: Thank you for this important point. Table 3 is our main evidence here: bounded composition is evaluated separately on **Easy**, **Medium**, and **Hard** proxy inspirations, and the gains remain broadly consistent across all three tiers. This suggests that the model is not simply overfitting to a specific region of the SPECTER2 similarity space, but is learning more robust composition under varying levels of retrieval mismatch.
>
> **Q3**: Training data is generated by R1-DISTILLED-QWEN-32B and filtered by a rubric-based evaluator (also LLM-based). While common, this could amplify model-specific biases.
>
> **A3**: Thank you for this important point. We agree that teacher-generated data and LLM-based rubric filtering can introduce or amplify model-specific biases. At the same time, this teacher-and-filter pipeline is a common practical recipe in reasoning post-training and distillation. In this paper, our primary goal is to study whether the training of \(P(h\mid b)\) can be made tractable through decomposition, hierarchical search, and bounded composition. We therefore adopt this standard setup as a practical training pipeline, while acknowledging that bias introduced by the teacher and evaluator is an important issue for future work.

---

> > ### Author Rebuttal · Reviewer_7pxv · 2026-04-04
> >
> > Thank you for the thoughtful rebuttal.

---

### Official Review · Reviewer_7SjB · 2026-03-13

**Soundness:** 3
**Presentation:** 3
**Significance:** 4
**Originality:** 3
**Overall Recommendation:** 5
**Confidence:** 4

**Summary:**

The paper proposes MOOSE-STAR, a framework for training LLMs to directly model P(h|b), the conditional probability of generating a scientific hypothesis from a research background. It identifies the combinatorial intractability of end-to-end training as a key limitation and addresses it through decomposed sequential training, bounded composition, motivation-guided hierarchical search, and a large-scale dataset TOMATO-STAR of 120,000 decomposed papers. Experiments suggest MOOSE-STAR achieves continuous test-time scaling while brute-force sampling saturates early.

**Compliance With Llm Reviewing Policy:**

Affirmed.

**Final Justification:**

The authors have addressed all my concerns.

**Key Questions For Authors:**

1. In Section 5.4, the hierarchical search complexity is stated as O(logN) in the best-case scenario. Could you provide an analysis or empirical estimate of the average-case complexity when accounting for realistic routing errors?
2. Could you provide the HC pass rate breakdown across k=1, 2, 3 steps (analogous to Table 5) to allow a direct comparison?
3. Tables 2 and 3 show that bounded composition data improves performance even under perfect ground-truth inspiration inputs. Could you provide explain why this occurs?

**Limitations:**

The paper include the Impact Statement that addresses potential negative societal impacts. But the discussion of technical limitations is lacking in the main text. Adding a brief paragraph explicitly acknowledging methodological boundaries would strengthen the paper.

**Strengths And Weaknesses:**

Strengths:
1. The paper targets a highly important and largely unexplored problem: the direct training of P(h|b), and further provides empirical scaling laws for this process, both of which are valuable contributions to the community
2. The work demonstrates clear originality in approaching P(h|b) training via decomposed subtasks, and both the hierarchical search strategy at inference time and the bounded composition training are novel and well-suited to the problem setting.
3. It is technically sound that the logical progression from formally identifying the complexity barrier to proposing specific, testable solutions is well-reasoned, and the experimental results generally support the claims.
4. The proposed TOMATO-STAR, comprising 120,000 decomposed papers spanning biology, chemistry, and cognitive science produced at substantial computational cost, is a meaningful standalone contribution that will benefit future research in this area.
5. It is clearly written with a well-organized structure that makes the narrative easy to follow


Weaknesses:
1. The claim of reducing complexity from O(N^k) to O(log N) deserves more rigorous delineation. Specifically, the manuscript should explicitly separate the fact that the reduction to a linear O(k*N) is achieved through the decomposed training framework , whereas the further reduction to O(log N) relies heavily on the test-time hierarchical retrieval routing perfectly.
2. While the paper's primary novelty is operationalizing the core probabilistic decomposition for training rather than inference, the core mathematical decomposition relies on a previously established formula from MOOSE-Chem(Yang et al., 2025b).
3. The discussion surrounding the tables (e.g. table2,3,4) are quite concise, forcing readers to spend additional time to interpret the data and grasp the findings. And the captions for tables are placed below the tables rather than above, which does not conform to style guidelines.

---

> ### Author Rebuttal · Authors · 2026-03-31
>
> We are grateful for the detailed comments.
>
> **Q1**: The manuscript should explicitly separate the fact that the reduction to a linear O(k*N) is achieved through the decomposed training framework, whereas the further reduction to O(log N) relies on hierarchical search design (in the best scenario).
>
> **A1**: Thank you for this helpful suggestion. We agree, and we have revised the manuscript to make this distinction more explicit. In particular, we now clarify that the reduction to linear \(O(kN)\) comes from the decomposed sequential training framework, while the further reduction to \(O(\log N)\) is enabled by hierarchical search in the best-case scenario.
>
> **Q2**: The core mathematical decomposition relies on a previously established formula from MOOSE-Chem.
>
> **A2**: We agree that the core decomposition formula originates from MOOSE-Chem. Our contribution here is different: while MOOSE-Chem proposed the formula and used it for inference, this paper shows how to operationalize it for the training of \(P(h\mid b)\), thereby turning a previously theoretical decomposition into a concrete training recipe.
>
> **Q3**: The discussion surrounding the tables (e.g. table2,3,4) are concise; captions for tables are placed below the tables rather than above.
>
> **A3**: Thank you. We agree, and we have revised the manuscript accordingly. Specifically, we moved the table captions above the tables and added more discussion to better interpret the results in Tables 2, 3, and 4.
>
>
> **Q4**: In Section 5.4, the hierarchical search complexity is stated as O(logN) in the best-case scenario. What about the average-case complexity?
>
> **A4**: Thank you for this important question. In our current analysis, the \(O(\log N)\) result is intended as a best-case complexity under effective guidance from the IR model. The average-case complexity depends on how well the IR model can guide traversal toward the correct semantic branch. If the guidance is strong, the search remains much closer to the logarithmic regime; if the guidance is weak, the process can gradually degrade toward near-linear search. We have clarified this point in the revised manuscript.
>
>
> **Q5**: Could you provide the HC pass rate breakdown across k=1, 2, 3 steps (analogous to Table 5) to allow a direct comparison?
>
> **A5**: We thank the reviewer for this insightful suggestion. We now provide the HC pass rate breakdown by $k$ in a unified table below, enabling direct comparison with the brute-force results:
>
> | **Method** | **Steps (*k*)** | **Total Samples** | **Passed Samples** | **Pass Rate** |
> |:--|:--:|:--:|:--:|:--:|
> | **Brute-force** | 1 | 31,314 | 653 | 2.09% |
> | *P(h \| b)* | 2 | 67,000 | 90 | 0.13% |
> | | 3 | 12,000 | 0 | 0.00% |
> | | *Total* | *110,314* | *743* | *0.67%* |
> | **HC (per step)** | 1 | 34,147×1 | 15,932 | 46.66% |
> | *P(Δh \| i, b)* | 2 | 57,904×2 | 57,116 | 49.32% |
> | | 3 | 14,708×3 | 23,081 | 52.31% |
> | | *Total* | *195,301* | *96,879* | ***49.60%*** |
>
> This breakdown reveals a key structural insight: the HC per-step pass rate remains stable at ~47–52% regardless of $k$, in stark contrast to brute-force sampling where the pass rate collapses from 2.09% ($k$=1) to 0.00% ($k$=3).
>
> **Q6**: Tables 2 and 3 show that bounded composition data improves performance even under perfect ground-truth inspiration inputs. Could you explain why this occurs?
>
> **A6**: We think the most likely reason is that, even when the ground-truth inspiration is provided, it does not always express the core scientific concept in the clearest or most directly usable way. Training with bounded composition data encourages the HC model to reason more robustly over semantically nearby but imperfect inspirations, which in turn helps it better recover the underlying core concept rather than relying on exact surface-form matching. As a result, this robustness can also improve performance under perfect ground-truth inspiration inputs.
>
>
> **Q7**: Adding a limitation section
>
> **A7**: Thank you for the suggestion. We agree, and we have added a dedicated **Limitations** section in the revised manuscript. In particular, we now explicitly discuss several current limitations of MOOSE-Star, including: (1) the current evaluation is still retrospective rather than fully forward-looking, (2) the \(O(\log N)\) complexity is a best-case result that depends on effective IR guidance, and (3) the efficiency of hierarchical search depends on the quality of the embedding space and retrieval design. We believe making these points explicit improves the clarity and balance of the paper.

---

> > ### Author Rebuttal · Reviewer_7SjB · 2026-04-02
> >
> > The authors have addressed all my concerns.

---

### Official Review · Reviewer_z3FD · 2026-03-13

**Soundness:** 3
**Presentation:** 3
**Significance:** 3
**Originality:** 4
**Overall Recommendation:** 5
**Confidence:** 4

**Summary:**

The article attempts to explore a major problem: the mathematical intractability of directly training Large Language Models (LLMs) to model the generative reasoning process of scientific discovery, specifically the conditional probability $P(h|b)$. The authors explore the central problem of combinatorial complexity ($O(N^k)$) that arises when an LLM must retrieve and compose a sequence of latent inspirations from a vast, global knowledge base. To break this barrier, the authors propose MOOSE-STAR, a unified framework that decomposes the objective into Inspiration Retrieval (IR) and Hypothesis Composition (HC). They introduce Hierarchical Search to reduce retrieval complexity to $O(\log N)$ and Bounded Composition to train the model to remain robust against imperfect, noisy retrievals. Additionally, a Motivation Planning step is used to dynamically prune the search space. The authors also release TOMATO-STAR, a large-scale dataset of 120,000 decomposed papers used to train this framework.

**Compliance With Llm Reviewing Policy:**

Affirmed.

**Final Justification:**

The authors have satisfactorily addressed the review concerns. The timestamped predictions provide a valuable record for systematic future comparison and will help assess the paper's long-term impact. While some limitations remain, the contributions are significant and clearly articulated. I recommend acceptance.

**Key Questions For Authors:**

1.  could the authors provide a case where MOOSE-STAR generates a hypothesis $h$ by retrieving inspirations $i$ from a knowledge base that predates the ground-truth paper?

**Limitations:**

yes

**Strengths And Weaknesses:**

## Strength
   1. The paper provides a formal analysis of why end-to-end hypothesis generation is hard ($O(N^k)$complexity) and proposes a principled decomposition into sequential subtasks.
   2. By training the Hypothesis Composition module on proxy inspirations within a semantic tolerance radius, the architecture becomes robust to the inevitable noise of practical retrieval systems. The authors thoughtfully stratify these proxies into Easy, Medium, and Hard tiers based on cosine similarity.
   3.  the dataset OMATO-Star, a dataset comprising 120,000 structurally decomposed scientific papers, is valuable for further research in this community.

## Weakness
   1. The Hierarchical Search relies entirely on an offline, static clustering of SPECTER2 embeddings. The methodological soundness of this component depends heavily on the assumption that SPECTER2's representation space perfectly aligns with the logical pathways needed for novel scientific discovery.
   2. While the paper demonstrates strong test-time scaling, the evaluation focuses strictly on retrospective discovery.  The effectiveness of the Motivation Planning strategy in generating truly unprecedented, forward-looking scientific theories remains unproven.

---

> ### Author Rebuttal · Authors · 2026-03-31
>
> Thank you for the insightful feedback.
>
> **Q1**: Hierarchical Search depends heavily on the assumption that SPECTER2's representation space perfectly aligns with the logical pathways needed for novel scientific discovery.
>
> **A1**: We agree that the embedding space matters for building an efficient hierarchical search tree. However, it does not need to perfectly align with the true logical pathways of scientific discovery to be useful. As shown in Table 4, hierarchical search already outperforms the strong baseline (tournament search), indicating that a meaningful semantic clustering is sufficient for search-tree construction. Since SPECTER2 is trained on scientific literature, it is a reasonable choice here. We also agree that stronger embeddings could further improve search efficiency, and view this as an important future direction.
>
> **Q2**: While the paper demonstrates strong test-time scaling, the proposed method has not yet predicted a hypothesis that predates the ground-truth paper.
>
> **A2**: We appreciate this important point. We address it from two perspectives.
>
> **(1) Timestamped predictions for future verification.**
> MOOSE-Star has only recently been completed, so strict temporal verification requires future publications. To support this, we have publicly released \~3k predicted hypotheses across \~1.5k research questions at `https://anonymous.4open.science/r/MOOSE-Star-Hypotheses-DA26`. These predictions provide a timestamped record for systematic future comparison.
>
> **(2) Quantitative evidence on temporally held-out papers.**
> While awaiting real-world validation, we also evaluate on papers published in October 2025, using a strict temporal split to ensure no data contamination. We define a prediction success as: **(i)** hierarchical search retrieves the ground-truth inspiration within top-50 of a 3,035-paper inspiration corpus, and **(ii)** the composed hypothesis reaches M3 score of at least 6. We evaluate 200 temporally held-out test samples. For clarity, in the second column below, **GPT-5.4 is used only as the HC executor within the MOOSE-Star framework after retrieval**.
>
> | HC Score | MS-HC-7B | GPT-5.4 as HC |
> |---|---:|---:|
> | >= 6 | 12.5% | 22.5% |
> | >= 7 | 4.0% | 20.5% |
> | >= 8 | 0.5% | 16.0% |
> | >= 9 | 0.0% | 10.5% |
> | >= 10 | 0.0% | 8.5% |
> | >= 11 | 0.0% | 4.0% |
> | >= 12 | 0.0% | 2.0% |
>
> These results do not replace future forward-looking validation. However, they show that on genuinely unseen papers, MOOSE-Star can already retrieve the correct inspiration within a small top-ranked subset of the corpus and, once retrieval succeeds, compose high-quality hypotheses for a meaningful fraction of cases.
>
> **Q3**: Could the authors provide a case where MOOSE-STAR generates a hypothesis by retrieving inspirations from a knowledge base that predates the ground-truth paper?
>
> **A3**: The following MOOSE-STAR generated hypothesis, though not predate, ensures no data contamination.
>
> **Paper:** *TREM2 Impedes Recovery After Spinal Cord Injury by Regulating Microglial Lysosomal Membrane Permeabilization-Mediated Autophagy* (*Cell Proliferation*, October 2025)
>
> **Question:** How to improve microglial survival and recovery after spinal cord injury via lysosomal integrity and autophagy regulation?
>
> MOOSE-Star retrieved the ground-truth inspiration from a 3k inspiration corpus at **Rank 1**: *TFEB links autophagy to lysosomal biogenesis* (*Science*, 2011).
>
> (Below hypotheses made more concise to fit the space limit)
>
> **Generated Hypothesis** (GPT-5.4 as HC executor, M3 score: 12/12) :
>
> - **Motivation:** Previous method identifies lysosomal membrane damage as a cause of microglial death after SCI, but not how surviving microglia rebuild degradative capacity. The inspiration paper suggests that TFEB is a master regulator that induces both lysosomal biogenesis and the autophagy machinery.
> - **Mechanism:** After SCI, microglia undergo oxidative, inflammatory, and metabolic stress that damages lysosomes. TFEB can restore the lysosomal-autophagy system by inducing lysosomal biogenesis, autophagosome formation, fusion, and degradation capacity.
> - **Methodology:** Test whether TFEB activation is impaired after SCI, whether it predicts lysosomal recovery, and whether it causally rescues autophagy flux, microglial survival, and recovery.
>
> **Ground Truth:**
>
> - **Motivation:** The specific limitation addressed is the lack of understanding of how to enhance autophagy in microglia after SCI.
> - **Mechanism:** TFEB is a master regulator of lysosomal biogenesis and autophagy; under stress, TFEB translocates to the nucleus and activates genes for lysosomal enzymes and autophagy-related proteins.
> - **Methodology:** Test TFEB nuclear translocation and downstream autophagy markers after SCI.
>
> This example shows that, on a genuinely unseen October 2025 paper, MOOSE-Star can retrieve a key inspiration from much earlier literature and use it to generate a hypothesis aligned with the actual scientific contribution.

---

> > ### Author Rebuttal · Reviewer_z3FD · 2026-04-01
> >
> > Thank you for your responses and the additional details. I believe the timestamped predictions provide a valuable record for systematic future comparison and will help assess the long-term impact of the paper.

---

### Decision · Program_Chairs · 2026-04-30

**Decision:**

Accept (regular)

**Comment:**

This paper identifies a fundamental barrier to training LLMs for scientific discovery: directly modeling P(hypothesis|background) is combinatorially intractable (O(N^k)) due to the need to retrieve and compose 'inspirations' from a large knowledge base. The authors propose MOOSE-Star, a framework that breaks this barrier through decomposed subtask training, motivation-guided hierarchical search reducing retrieval to O(log N) in the best case, and bounded composition for robustness to retrieval noise, and additionally release TOMATO-Star, a dataset of 120,000 structurally decomposed papers.

**Strengths.** The problem is well-motivated and formally grounded: the tractability analysis provides a principled justification for the architectural choices. The decomposition into inspiration retrieval and hypothesis composition is technically sound, and bounded composition's tiered training strategy (easy/medium/hard proxy inspirations) is a thoughtful solution to a real practical bottleneck. The TOMATO-Star dataset, which required extensive computational resources to generate, is a valuable standalone contribution for the AI-for-Science community. The test-time scaling results, showing continuous improvement where brute-force sampling saturates, are among the paper's strongest empirical contributions.

**Concerns.** Reviewers noted that the core probabilistic decomposition builds on prior work (MOOSE-Chem), and that the evaluation relies heavily on LLM-based judges (Gemini) rather than human expert assessment. The O(log N) complexity claim pertains to the best case and its separation from average-case behavior warranted clarification. Evaluation is retrospective, leaving forward-looking generalization unproven.

**Rebuttal.** Two of four reviewers raised their scores. Reviewer 5wGb upgraded from 4 to 5 after the authors provided evidence that their automated evaluation pipeline correlates with human expert judgment, directly addressing the main concern. Reviewer z3FD upgraded from 4 to 5 after the authors clarified the timestamped prediction setup and addressed concerns about retrospective evaluation. The remaining reviewers maintained positive scores.

**Bottom line.** A technically grounded contribution to a largely unexplored problem, with a strong dataset release which is likely to be useful for the community, convincing scaling results, and a reviewer set that converged toward strong acceptance after rebuttal.